# Normalization governs attentional modulation within human visual cortex

Ilona M. Bloem [1,2]* & Sam Ling[1,2]

Although attention is known to increase the gain of visuocortical responses, its underlying neural computations remain unclear. Here, we use fMRI to test the hypothesis that a neural population's ability to be modulated by attention is dependent on divisive normalization. To do so, we leverage the feature-tuned properties of normalization and find that visuocortical responses to stimuli sharing features normalize each other more strongly. Comparing these normalization measures to measures of attentional modulation, we demonstrate that sub-populations which exhibit stronger normalization also exhibit larger attentional benefits. In a converging experiment, we reveal that attentional benefits are greatest when a subpopulation is forced into a state of stronger normalization. Taken together, these results suggest that the degree to which a subpopulation exhibits normalization plays a role in dictating its potential for attentional benefits.

[1] Department of Psychological and Brain Sciences, Boston University, Boston, MA 02215, USA. [2] Center for Systems Neuroscience, Boston University, Boston, MA 02215, USA. *email: ibloem@bu.edu

Neural processing is surprisingly efficient. Although our environment is brimming with information, our cognitive system adeptly regulates competition between neural representations—all vying for visual awareness. A growing body of evidence suggests that this is made possible, in part, by recruiting a seemingly ubiquitous neural computation, known as divisive normalization, which can regulate the relative strength between competing representations[1–4]. Under normalization, the response to a stimulus is modulated by the summed activity generated by the stimulus itself, along with pooled neighboring responses. This computation crucially supports a number of functions, including regulating the dynamic range of neural responses[2,3,5,6]. Models of divisive normalization have long served as cornerstone principles for computational accounts of early vision, and generalize to a variety of other sensory modalities[3,7,8] and cognitive processes[5,9–14]. Interestingly, normalization is also believed to be modulated by contextual influences, whereby visual features that are similar tend to normalize each other more than those that are dissimilar[10,15–18]. This feature-tuned aspect has been theorized to play an active role in reducing redundant sensory information[17,19–22]. The image properties in our visual environment are comprised of statistical biases, whereby neighboring features belonging to the same object are most likely to be similar. Feature-tuned normalization incorporates these inherent dependencies, acting as a form of neural information compression by deprioritizing the processing of redundant representations[17,21,22], thereby aiding in the segregation of a figure from its background.

While tuned normalization may play a role in the bottom-up enhancement of potentially relevant information in a visual scene, we ultimately rely on top-down attentional systems to selectively enhance a small subset of that information for prioritized processing, from moment to moment. One of the most well-documented ways that attention enhances relevant information is by increasing the gain, or 'strength', of the behavioral[23–26] or neural response[5,27–30]. Interestingly, prominent computational models have theorized that normalization and attention are tightly linked, whereby attentional modulation within visual cortices is dependent on divisive normalization[5,31,32]. These models propose that attention can alter the balance between the stimulus activity and the summed activity of the normalization pool, weakening the current state of gain control, and thereby resulting in an increased neural response. Normalization accounts of attention have traditionally hinged on three key components: the locus of attention, the size of the stimulus, and the size of the attentional window[5,28,33]. However, while these models have proposed that the spatial extent of the 'attention field' could be feature selective[5,26,34], the suppressive drive itself is typically considered to be feature-agnostic, allowing an equal contribution of all information, regardless of feature similarities. The notion that attention modulation could additionally depend on a fourth component, incorporating the feature-selective nature of normalization, has some support in animal studies, with single-unit recordings in macaques suggesting that the contribution of tuned normalization can explain attention biases of competition between multiple stimuli within a receptive field[10].

In this study, we use functional magnetic resonance imaging (fMRI) to test the hypothesis that attention-driven modulation of the gain of responses within human visual cortex depends on the magnitude of tuned normalization. We approach this problem by first devising an efficacious, voxel-by-voxel population measure of the feature-tuned aspects of normalization within early visual cortex, during passive viewing. To do so, we exploit a phenomenon known as sub-additivity, a signature property of normalization wherein the population responses to images comprised of two superimposed stimuli fall short of the linear sum of the

response to each stimulus independently[35–40]. We demonstrate potent tuned normalization within human visual cortex: superimposed stimuli sharing the same features are more strongly normalized than stimuli that differ in their features. Armed with a population measure of feature-tuned normalization, we set out to test the hypothesis that attentional modulation is partially driven by tuned normalization. If normalization truly governs attentional modulation, we reason that attention-driven gain changes will be greater when a neural subpopulation within early visual areas exhibits stronger normalization. Indeed, in our second experiment we reveal that tuned normalization is tightly linked to an independent measure of attentional modulation. Leveraging population-wide heterogeneities in BOLD responses for both normalization and attention measures, we find that subpopulations that exhibit stronger normalization also exhibit larger attentional benefits. Finally, in a third converging experiment, we directly manipulate spatial attention, while simultaneously measuring population activity under different states of normalization. In doing so, we demonstrate that attentional benefits are greater when the population is put under stronger normalization. In sum, our results suggest that a neural population's capability for attentional benefits appears contingent upon normalization, whereby the degree to which a population can normalize itself results in greater potential for attentional modulation.

## Results

**Sub-additivity as a signature of tuned normalization.** We first set out to obtain a population measure of visuocortical responses under different states of normalization. Specifically, in addition to a well-known untuned, feature-agnostic component, does sub-additivity show a signature of a tuned, or feature-selective component? We leveraged the fact that population responses to images comprised of superimposed visual stimuli are not simply the linear sum of the response to each stimulus independently[35–38]. Instead, the response typically exhibits a property known as sub-additivity—a phenomenon nicely captured by contrast normalization. This is believed to emerge due to the compressive nature of normalization, which acts to nonlinearly limit the overall response to the stimuli. To assess the influence of tuned and untuned normalization on population responses within human visual cortex, we leveraged an fMRI noise-masking technique[24,36,38], which allowed us to test the degree to which BOLD responses within early visual cortex exhibit sub-additivity, depending on stimulus feature similarity. To tap into the sub-additive nature of tuned normalization, we constructed stimuli that were composed of linearly summed pairs of oriented bandpass-filtered noise gratings (outer diameter 15°; inner diameter 3°; at 50% Michelson contrast; spatial frequencies between 2 and 3 cycles/°; orientation bandwidth of 10°). Importantly, images were constructed using pairs of stimuli combined in either an orthogonal (different features) or a collinear configuration (similar features; Fig. 1a). We measured BOLD responses to these collinear and orthogonal stimuli configurations in separate blocks during an fMRI session, while participants performed a demanding task at fixation, finding targets in a rapid letter stream, and ignoring the stimuli presented in the periphery (Supplementary Fig. 1). In addition, in a separate set of scans we measured the BOLD response to both individual components (45° & 135°), in separate blocks, that comprised the overlaid stimuli, and summed the responses for each component in order to create a hypothetical additive sum. Note, we did not find a difference in the evoked responses to individual components when oriented radial or tangential from fixation (Supplementary Fig. 2). The sub-additive deviation from the hypothetical sum for both the collinear and orthogonal configurations served as our measure of untuned

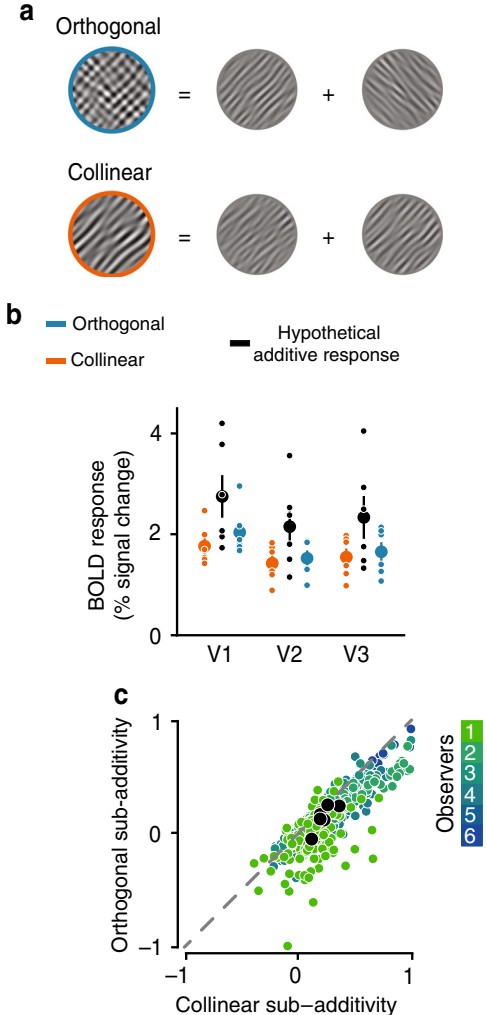

**Fig. 1 Sub-additivity as a measure of tuned normalization. a** Schematic of stimuli used to measure sub-additivity. Two oriented stimuli components (outer diameter 15°; inner diameter 3°, at 50% Michelson contrast, spatial frequencies between 2 and 3 cycles/°; orientation bandwidth of 10°) were linearly summed in either an orthogonal (top; blue) or a collinear (bottom; orange) configuration, resulting in a full contrast stimulus. Stimuli are modified for illustrative purposes. **b** Average BOLD responses across observers for orthogonal (blue) and collinear (orange) configurations. A hypothetical additive response (black) was created by summing the average response evoked by both individual stimulus components (45° & 135°). Dots represent individual observers; $N = 6$; error bars denote ±1 S.E.M. **c** Voxel-wise relationship of the deviation from additivity for both stimuli configurations in V1. BOLD responses were normalized for each participant. Small colored dots indicate individual voxels; larger black dots represent the whole ROI average per observer.

normalization, while comparing the difference between responses evoked by the two overlaid configurations served as our measure for an additional tuned component.

Orthogonal and collinear stimuli configurations both exhibited sub-additivity across early visual areas (V1–V3), with the measured BOLD response of both stimulus configurations being lower than the hypothetical additive sum (repeated measures ANOVA orthogonal sub-additivity: main effect of additivity $F(1,15) = 16.39$, $p = 0.001$, $\eta_p^2 = 0.522$, main effect for visual area and the interaction, $p > 0.3$; repeated measures ANOVA collinear sub-additivity: main effect of additivity $F(1,15) = 22.84$, $p < 0.001$, $\eta_p^2 = 0.604$, main effect for visual area and the interaction, $p > 0.4$;

Fig. 1b, Supplementary Fig. 1). Furthermore, the responses demonstrated robust feature-tuned normalization as well, whereby stimuli comprised of collinear orientations were more sub-additive, and thus more strongly normalized, than stimuli that contained orthogonal orientations (repeated measures ANOVA interaction effect $F(2,15) = 7.85$, $p = 0.005$, $\eta_p^2 = 0.511$; tuned normalization post hoc analysis two-sided paired $t$-test; V1: $t(5) = 6.00$, $p = 0.0057$, $d = 2.44$, V2: $t(5) = 3.82$, $p = 0.0370$, $d = 1.56$, and V3: $t(5) = 3.44$, $p = 0.0551$, $d = 1.42$, Bonferroni corrected). While BOLD responses to either stimuli configuration across visual areas were fairly consistent in the degree to which they exhibited sub-additivity (untuned normalization), the magnitude of the feature-tuned aspect of normalization decreased in strength along the visual hierarchy (one-way ANOVA across visual areas: $F(2,17) = 7.85$, $p = 0.005$, $\eta_p^2 = 0.511$). This orientation-tuned aspect of normalization was strongest within primary visual cortex—a region shown to be most precisely tuned to orientation content[41–43], and became less apparent as we moved up the visual hierarchy, consistent with a shift in the preferred feature space. We next explored the degree of dependency, from voxel-to-voxel, of the deviation from additivity for both orthogonal and collinear stimuli configurations within V1. Although there was heterogeneity in the magnitude of sub-additivity between voxels within a region, comparing the difference between the hypothetical additive sum with both collinear and orthogonal stimulus configurations revealed a consistent pattern, with the collinear configuration exhibiting larger sub-additivity compared with the orthogonal configuration —an effect that was highly reliable for the majority of voxels within V1, but decreased in V2 and V3 (Fig. 1c, Supplementary Figs. 3, 4).

Importantly, the differences in BOLD responses evoked by the two stimuli configurations were not driven by differences in the image statistics, adaptation (see Supplementary Fig. 5), nor could they be explained by basic first-order visual response properties. A Fourier analysis confirmed that while the orientation content between the stimuli configurations differed, the overall power was comparable (Fig. 2a). Furthermore, a V1-based energy detection model[44–49] that only incorporated untuned divisive normalization also fell short of accounting for these results. In this model, we estimated the amount of contrast energy each class of images evoked by applying a linear Gabor wavelet decomposition that described tuning along the dimensions of space, phase, orientation, and spatial frequency[44,45] (Fig. 2b). Five hundred unique images for both collinear and orthogonal stimuli configurations were passed through the model, which resulted in a measure of contrast energy for each image, contained in quadrature wavelet pairs. After combining all wavelets across space and spatial frequency scales, a measure of contrast energy evoked by each orientation channel remained. The model output then underwent divisive normalization, effectively acting as a contrast gain control operator[2,6,50]. A bootstrap analysis indicated that there is no difference between the two simulated stimulus energy distributions, indicating that these images have the same stimulus energy when applying untuned normalization (95% confidence interval = [−0.026, 0.019]; Fig. 2c). Importantly, however, a difference in stimulus energy was observed when we incorporated an orientation-tuned component into the normalization model (bootstrapped 95% confidence interval = [0.362; 0.4035]; Fig. 2d), indicating that the observed differences in sub-additivity of the BOLD responses between the two stimuli configurations can be driven by tuned normalization.

**Attentional modulation is related to tuned normalization.** Leveraging our ability to measure feature-tuned normalization

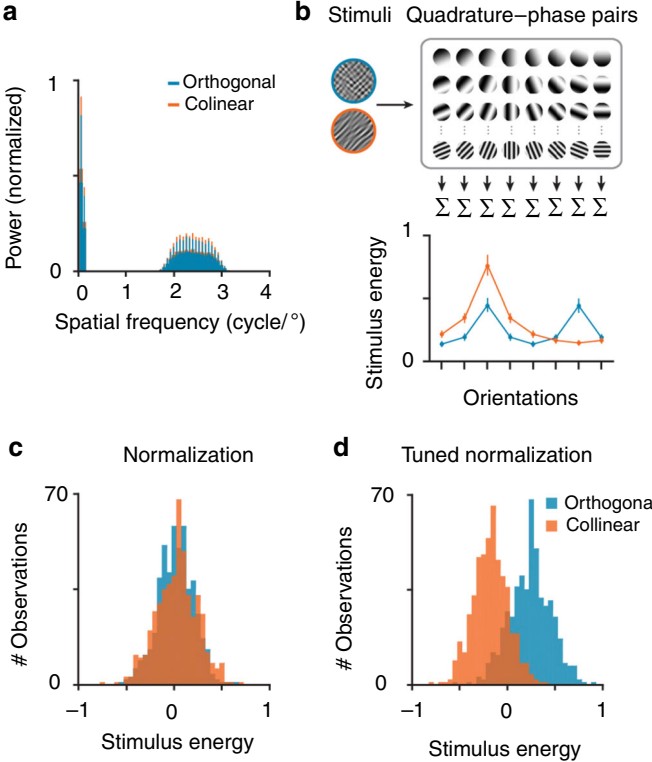

**Fig. 2 Image statistics between collinear and orthogonal stimuli did not differ. a** Average normalized power obtained with a 2D Fourier transform of 500 images within each stimulus configuration (collapsed across phase). Error bars denote ±1 S.E.M. **b** Energy detector model. Stimuli were convolved with a set of Gabor filters across space (centered on every pixel of the image), phase (two phases to create a quadrature pair), spatial frequency (36 scales, ranged between 0.5–4 cycles/°), and orientations (eight orientation channels). The quadrature-phase pairs were squared, summed, square-rooted, and normalized to 1, to generate a complex-cell response at the eight orientation channels for each image, see Methods. Overall stimulus energy was demeaned across all images of both stimuli configurations and normalized by its maximum value, mainly for illustrative purposes. Error bars denote ±1 S.E.M. **c** To account for contrast saturation, divisive normalization was applied to the total stimulus energy obtained from the filter outputs, and we combined all orientation channels to obtain one stimulus energy value per image. As is evident, the distributions of stimulus energy for both stimuli configurations are highly comparable when divisive normalization is applied. **d** Incorporating a tuned component to the normalization term results in a separation of the energy distributions in which collinear stimuli have a relatively lower stimulus energy, see Methods.

within human visual cortex, we then set out to test our main hypothesis: Does attention boost information processing by modulating divisive normalization? Our previous experiment demonstrated that collinear stimulus configurations were more sub-additive, as they evoked lower BOLD responses compared with orthogonal stimuli. Importantly, while responses to each stimulus configuration varied substantially, this difference between collinear and orthogonal configurations was consistent for almost all voxels within an area (Fig. 1c, Supplementary Fig. 4). In a second experiment, we examined the degree of voxel-wise dependency between this measure of tuned normalization and an independent measure of attention modulation. If normalization and attentional modulation interact, we predicted that those subpopulations that are more strongly normalized should also exhibit the highest potential for attentional modulation.

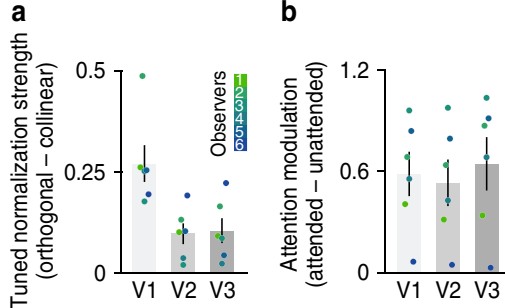

**Fig. 3 Measures of tuned normalization strength and attention modulation. a** Tuned normalization strength reflects the difference between BOLD responses evoked by orthogonal and collinear stimuli blocks (% signal change). The orthogonal configuration elicited larger BOLD responses compared with the collinear configuration across V1–V3. **b** Attention modulation reflects the difference in BOLD response between attended and unattended stimuli blocks (% signal change). We observed large attention modulation across V1–V3. Error bars denote ±1 S.E.M.; $N = 6$; colored dots indicate individual observers.

Tuned normalization strength was quantified as the difference between the mean BOLD responses to orthogonal and collinear stimuli blocks. Collinear stimuli evoked weaker BOLD responses compared with orthogonal stimuli, resulting in a positive difference for all regions of interest (Fig. 3a). To measure attentional modulation, we assessed BOLD responses while participants viewed orientation bandpass-filtered noise gratings (outer diameter 15°; inner diameter 3°; at 50% Michelson contrast; spatial frequencies between 2 and 3 cycles/°; orientation bandwidth of 10°). While viewing these stimuli, participants were asked to either covertly attend toward the stimulus, performing a fine orientation discrimination task (Attended condition), or perform a demanding task at fixation, which drew attention away from the oriented stimulus (Unattended condition). Note that the visual stimulation was identical in both conditions, with the only difference being the task observers performed (Supplementary Fig. 6). Attention modulation was defined as the difference between BOLD responses to attended and unattended stimuli blocks. Consistent with previous findings[24,51–53], striate and extrastriate cortex exhibited robust attentional modulation (Fig. 3b; repeated measures ANOVA: main effect of attention $F(1,15) = 50.38$, $p < 0.001$, $\eta_p^2 = 0.771$, while main effect for visual area and the interaction did not reach significance, $p > 0.4$).

While these results reflect the average response across the entire region of interest (ROI), the magnitude of tuned normalization strength varied substantially from voxel-to-voxel within each visual area, suggesting heterogeneity across the population (Fig. 1c). Leveraging this population-wide heterogeneity in neural responses in both attentional modulation and tuned normalization measures, our results revealed that subpopulations that exhibit the strongest tuned normalization also possess the greatest attentional benefits, most strongly within V1 and V3 (Fig. 4; one-way ANOVA on Fisher-Z transformed Spearman correlations; $F(2,17) = 3.326$, $p = 0.064$, $\eta_p^2 = 0.307$; post hoc two-sided one sample $t$-tests V1: $t(5) = 4.30$, $p = 0.0231$, $d = 1.76$, V2: $t(5) = 2.96$, $p = 0.0944$, $d = 1.21$, and V3: $t(5) = 5.14$, $p = 0.0109$, $d = 2.10$, Bonferroni corrected). To explore the spatial distribution of these effects, we examined the correspondence between the retinotopic preference and attentional modulation or tuned normalization strength measures. Using pRF mapping procedures[54,55], we estimated the preference for spatial position for every voxel, allowing us to assess potential biases in both measures of interest, based on retinotopic preference. There did

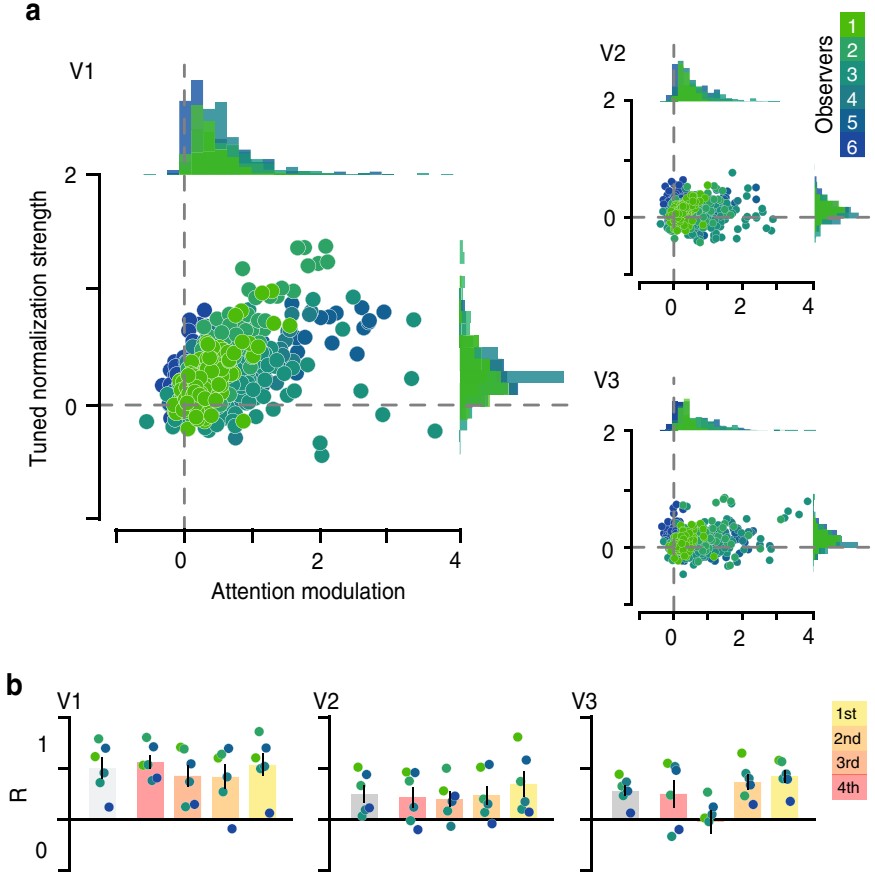

**Fig. 4 Attentional modulation as a function of tuned normalization strength. a** A tight relationship between tuned normalization strength and attentional modulation is evident for the top 25% selected voxels for each observer, see Methods. Dots illustrate individual voxels within an area, colors represent a unique participant. **b** Spearman correlations were computed for each observer, gray bars represent the mean correlation across observers, while the colored bars represent the correlations when the voxel selection is broken down into four bins based on the independent visual localizer (red: bottom 25%, yellow: top 25%). All Spearman correlations were transformed into a Fisher-Z-statistic to allow for statistical comparisons between observers, see Methods. Error bars denote ±1 S.E.M.; N = 6; colored dots illustrate individual observers.

not appear to be any strong clustering of either of our two measures across spatial position. This suggests that while both attentional modulation and normalization strength share a tight relationship, this is not driven by a systematic coarse-scale topographical organization (see Supplementary Fig. 7). Furthermore, to ensure that our results reflected a true relationship between normalization and attention, rather than being driven by differences in spurious factors, such as the signal-to-noise ratio (SNR), our analyses were restricted to the top 25% most visually responsive voxels within V1–V3, selected by ranking the significance obtained from a standard GLM analysis of an independent functional localizer. Note that the relationship persists when all voxels within a respective region are used in the analyses (one-way ANOVA on Fisher-Z transformed Spearman correlations; $F(2,17) = 2.78$, $p = 0.094$, $\eta_p^2 = 0.270$; post hoc two-sided one sample $t$-tests V1: $t(5) = 5.30$, $p = 0.0096$, $d = 2.16$, V2: $t(5) = 3.42$, $p = 0.0567$, $d = 1.40$, and V3: $t(5) = 4.66$, $p = 0.0166$, $d = 1.90$, Bonferroni corrected). In addition, we examined whether the relationship between tuned normalization strength and attentional modulation was still evident even when we broke down our stringent voxel selection into four bins, according to ranked goodness of fit of responses to the visual localizer (Fig. 4b). While the observed correlation within V1 and V2 was not driven by differences in the goodness of fit of the localizer scans, as a similar relationship persisted in each bin, within V3 the correlation did seem driven by voxels that had a better fit

(one-way ANOVA of Fisher-Z transformed Spearman correlations across bins; V1: $F(3,20) = 0.52$, $p = 0.672$, $\eta_p^2 = 0.073$; V2: $F(3,20) = 0.62$, $p = 0.612$, $\eta_p^2 = 0.085$; and V3: $F(3,20) = 3.97$, $p = 0.023$, $\eta_p^2 = 0.373$). The less pronounced relationship in extrastriate cortex is likely driven by the reduced heterogeneity and overall magnitude of tuned normalization strength observed in these visual areas (Fig. 4, Supplementary Fig. 8). Taken together, leveraging the heterogeneity of population responses for attentional modulation and tuned normalization strength, our results reveal a tight link between these two measures, which was strongest in primary visual cortex, suggesting that a neural subpopulation's potential to increase its attentional gain is dependent on its tuned normalization strength.

**Tuned normalization modulates spatial attention.** To provide converging evidence in support of the underlying relationship between tuned normalization and attention, we carried out an additional experiment, wherein we directly assessed whether attentional modulation is greater when the population response is put under a state of stronger normalization. To do so, we measured BOLD responses for the overlaid stimuli configurations in separate blocks, similar to those constructed in Experiment 1, while covert spatial attention was directed to either the left or right side of fixation. To leave enough headroom for an increased BOLD response with attention, we used a lower contrast stimulus

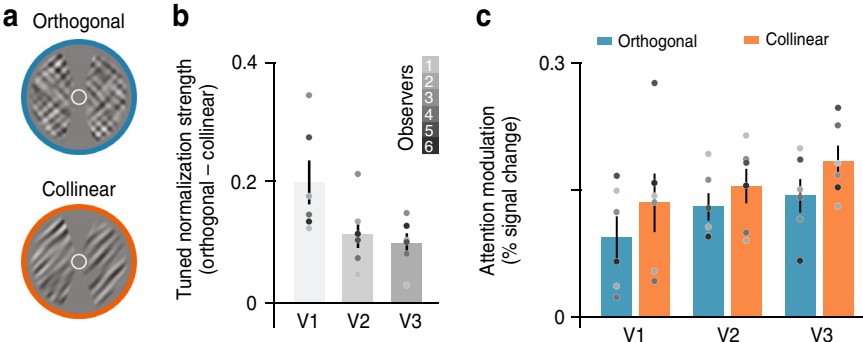

**Fig. 5 Tuned normalization modulates spatial attention. a** Stimuli were comprised of orientation bandpass-filtered noise gratings (outer diameter 15°; inner diameter 3°; spared midline; individual component was rendered at 25% Michelson contrast, resulting in a combined grating of 50% Michelson contrast), comparable to stimuli used for Experiment 1. Participants performed a demanding spatial attention task, detecting and discriminating a small gray Gaussian disk embedded in either the upper or lower visual field of the attended location (a cue presented throughout the block at fixation informed the participant to attend the left or right side of fixation). **b** Tuned normalization strength represents the difference between BOLD responses evoked by orthogonal and collinear stimuli blocks when attention was directed away (% signal change). The orthogonal configuration elicited larger BOLD responses compared with the collinear configuration across V1–V3. **c** Attention modulation reflects the difference between BOLD responses when spatial attention was either directed toward or away from the stimuli locations, for both collinear and orthogonal configurations. Error bars denote ±1 S.E.M., N = 6, gray dots illustrate different participants; stimuli are modified for illustrative purposes.

(individual components 25% contrast, resulting in a combined overlaid stimulus of 50% contrast). Participants performed a demanding probe detection task, detecting and discriminating whether a neutral gray Gaussian disk was embedded in the upper or lower visual field of the attended side of the screen, while maintaining fixation at the center of the screen (Fig. 5a, Supplementary Fig. 9, diameter probe 1.5°). Note that the targets could appear on either side of fixation, and only the cue at fixation would indicate where to attend. This experimental design allowed us to simultaneously measure BOLD responses for either configuration when attention was directed toward or away from the stimulus.

First, we assessed whether stimuli that share feature information (collinear configuration) yield stronger tuned normalization, compared with stimuli with dissimilar features (orthogonal configuration), when attention was directed to the opposite visual field. In agreement with the results of Experiment 1, we found strong tuned normalization that decreased in its strength across visual areas (Fig. 5b; repeated measures ANOVA interaction effect $F(2,15) = 4.116$, $p = 0.038$, $\eta_p^2 = 0.354$; tuned normalization post hoc analysis two-sided paired $t$-test; V1: $t(5) = 5.46$, $p = 0.008$, $d = 2.23$, V2: $t(5) = 4.84$, $p = 0.014$, $d = 1.99$, and V3: $t(5) = 5.80$, $p = 0.006$, $d = 2.32$, Bonferroni corrected; one-way ANOVA across visual areas $F(2,17) = 4.12$, $p = 0.038$, $\eta_p^2 = 0.354$). Having established that there is strong tuned normalization when attention is directed away, we then set out to test whether tuned normalization truly dictates the magnitude of attentional modulation. We hypothesized that the largest attentional effects would be evident when a neural population experiences stronger normalization, induced by the similarity between the features of the overlaid stimuli (i.e., attentional effects for collinear >orthogonal).

To quantify the magnitude of attentional modulation, we computed the difference between responses when attention was directed toward vs. away, from either the collinear or orthogonal stimuli configurations. We discovered that attentional effects were indeed the greatest for the collinear stimulus configuration, when neural responses were put under a stronger state of normalization (Fig. 5c; repeated measures ANOVA main effect of stimulus configuration $F(1,15) = 12.43$, $p = 0.003$, $\eta_p^2 = 0.453$; while main effect for visual area and the interaction did not reach significance, $p > 0.2$). These results provide direct evidence to

suggest that a more strongly normalized population is more susceptible to an attention-facilitated increase in gain.

## Discussion

Taken together, our results reveal that a neural population's capability for attentional benefits is tightly linked to feature-tuned normalization. The magnitude of attentional modulation depends on the degree to which a population has normalized its response, based on the degree of feature similarity within an image. In the first experiment, we utilized an efficacious method to probe orientation-tuned normalization of population responses within human early visual cortex. By superimposing stimuli that differed in their orientation content, we found that BOLD responses were lower for stimuli that matched in their visual features, compared with stimuli that were comprised of different features. In a second experiment, we found a tight voxel-wise relationship between this measure of tuned normalization strength and an independent measure of attention modulation, suggesting that attention optimizes information processing by modulating divisive normalization. Critically, in a third experiment, we provided direct converging evidence that the magnitude of attentional benefits depends on the degree to which a population has normalized its response; when a neural population was put under a stronger suppressive state, the largest attentional effects emerged.

Our results square with a normalization-based model of attention, which posits that attentional modulation arises through interactions with divisive normalization[5,14,31,32]. This model is the prevailing theory, to date, by which attention is believed to act upon neural responses. While previous work provided support for this model[1,28,33,34,56], our results extend the notion that normalization-driven properties of attention are feature selective[10,30,57]. The standard normalization model proposes that the spatial extent of an 'attention field' can reshape either relative to the stimulus size or to match a particular feature in order to modulate a population response, and suppression is considered to be feature-agnostic, acting independently from the selective properties computed within a respective region. However, divisive normalization is modulated by contextual influences, where feature similarity results in stronger normalization, a property the model currently does not account for. Extending the standard normalization model by incorporating a tuned component into the suppressive drive could potentially account for differences

between the two stimuli configurations (Fig. 2d and see the Methods section). Previous work has suggested that instead of incorporating a tuned suppressive component into the normalization model of attention, a more parsimonious description to explain differences in the magnitude of suppression is by allowing attention to be feature-selective[5,26,34]. While the notion of feature-based attention is well established[58,59], incorporating this into the normalization model does not account for the results we have presented here. In our study, we provide evidence that differences between the two stimuli configurations emerge even when they were unattended (Figs. 2d and 5b), suggesting that tuned inhibition can arise in the absence of any aid of top-down attentional feedback. Furthermore, in a third experiment we manipulated spatial attention, while holding factors such as stimulus size, attentional window size, and contrast relatively constant, in order to investigate the role that feature-tuned normalization has on attentional modulation (Fig. 5). We find that the magnitude of attention is greatest when stimuli match in their visual features. This finding is not readily explained by the normalization model of attention, and instead would need to be extended such that feature similarity results in a release from normalization with attention. Future research measuring the full neural contrast response function, and manipulating features such as the size and shape of the attentional window, will shed more light on precisely how this contribution of feature-tuning is best incorporated into the normalization model of attention.

Normalization has been proposed to be a canonical computation throughout cortex, and relies on several mechanisms which all serve to regulate the relative strength between neural representations. Two well-established mechanisms within early visual cortex are surround suppression and cross-orientation inhibition[60,61]. Surround suppression is characterized as the modulation of the neural response within the classical receptive field, as a result of the intensity of stimulation presented outside of the receptive field[22,62], while cross-orientation inhibition is the modulation of the neural response induced by presenting two superimposed oriented stimuli components within the classical receptive field[61,63,64]. Neuroimaging and psychophysical experiments cannot precisely target a single receptive field, and instead these methods measure population responses evoked by relatively large stimuli, with the spatial area typically spanning far beyond the receptive field of any individual neuron[65]. This likely makes the interactions arising from overlay stimuli, as used here, more analogous to surround suppression. While neuroimaging and psychophysical studies using a typical center-surround stimuli often report an attenuation of the response to the center stimulus[15,39,66], it is important to consider that this center does not correspond to any particular receptive field center. Instead, the center stimulus drives the response of a large population of neurons, of which only those neurons close to the border between the center and surround stimulus are likely to be attenuated[65,67]. In this study, we set out to optimize surround suppression by superimposing our stimuli configurations, presented full field (15° visual angle stimulus diameter). We hypothesized that by keeping the orientation of one of the components constant and manipulating the orientation of the second component, we can induce normalization more analogous to surround suppression within all neural populations with receptive fields falling within our stimulus bounds. The superimposed configurations indeed elicited the predicted population responses that one would expect from feature-tuned normalization[15,65,68,69], as we found lower BOLD responses for those configurations that matched in their orientation content, compared with configurations with orthogonal orientation information.

Feature-tuned normalization is suggested to play an active role in the efficient coding of natural stimuli[17,20–22,70,71]. Our visual environment is comprised of statistical biases between image features, whereby nearby edges have a higher probability to be co-oriented and belonging to the same contour, as compared with more distant edges[17]. A tuned normalization pool could perhaps incorporate these statistical dependencies by attenuating its strength where features match, leading to an effective boost of the responses at discontinuities, where features are no longer quite as co-aligned. While tuned normalization plays a role by prioritizing processing for salient items, potentially aiding the visual system in segregating figure from ground, we ultimately rely on top-down attentional systems to selectively enhance a small subset of that information for prioritized processing. Selective attention may interact with this figure/ground segregation process by selectively highlighting objects in the environment, which often are defined by their common feature properties.

## Methods

**Observers**. Six healthy adults participated in the first two experiments (3 male, mean age = 30), and seven adults (2 male, mean age = 28) participated in the third experiment. Five adults participated in all three experiments. All observers provided written informed consent, and had normal or corrected-to-normal vision. The Boston University Institutional Review Board approved the study. One observer who participated in the final experiment was excluded from further data analysis, based on consistent eye movements toward the cued spatial locations (eye-movement analysis revealed a mean deviation from fixation of >1°). A power analysis indicated that six subjects would be sufficient to detect the predicted attention effects. Indeed, this sample size is consistent with previous fMRI attention and vision studies[24,28,33,34,51,53,59,72].

**Apparatus and stimuli**. Stimuli were generated using Matlab (R2013a) in conjunction with the Psychophysics Toolbox[73,74], rendered on a Macbook Pro (OS X 10.7), and were displayed on a rear-projection screen (subtending ~21° × 16°) using a gamma-corrected projector. Participants viewed the display through a front surface mirror. Participants were placed comfortably in the scanner with their heads fixed, using padding to minimize head motion. Stimuli consisted of bandpass-filtered noise gratings (outer diameter: 15°; inner diameter: 3°; at 50% Michelson contrast). The bandpass filter spared only spatial frequencies between 2 and 3 cycles/°, orientation content centered at 45° or 135° (orientation bandwidth of 10°), and was smoothed in the Fourier domain to avoid Gibbs ringing artifacts.

**Tuned normalization experiment**. Stimuli were the linear combination of the stimuli described above. Superimposing these components created either orthogonal (45°/135° and 135°/45°) or collinear (45°/45° and 135°/135°) stimuli, which resulted in a doubling of the contrast to 100% Michelson contrast.

The two overlaid stimuli configurations were presented at 2 Hz (250 ms on, 250 ms off) for 14 s, where each stimulus presentation within a block consisted of unique random noise stimuli. Blocks (2 s cue; 14 s stimulus presentation) were pseudo-randomized over the course of a run, and interleaved with 16 s fixation periods. Throughout the experiment observers performed a demanding fixation task, finding targets in a rapid letter stream presented at fixation (5 Hz, letter size: 0.7°). During stimulus presentation blocks, target letters would appear with a probability of 30%, and participants reported whenever they detected a 'J' or a 'K' amongst distractor letters (Supplementary Fig. 1). Observers were capable of discriminating the target letters with high accuracy (mean = 0.93, S.E.M. = 0.02).

In addition to the overlaid stimuli, in separate runs we measured the BOLD response to an individual stimulus component (50% Michelson contrast), and doubled this obtained response to create a hypothetical additive sum. Oriented stimuli (45° & 135°) were presented at 2 Hz in separate blocks (2 s cue, 14 s stimulus presentation), and were interleaved with baseline periods. Observers performed the same fixation task as described above, reporting the presence of target letters embedded in a rapid letter stream presented at fixation (performance: mean = 0.88, S.E.M. = 0.04). Participants completed 5–10 fMRI runs; each run took 272 s to complete (8 stimulus blocks per run). In addition, a scan session included two visual localizer scans, in which a flickering and rotating contrast pattern was presented within the same aperture as the filtered noise stimuli (blocked presentation, 16 s on and off; 6 stimulus blocks per run).

**Attention modulation experiment**. To examine the voxel-wise relationship between tuned normalization and attention, we obtained a measure of attentional modulation within the same scan sessions as the previous experiment (n = 6). In this experiment, orientation stimuli (single component of the stimuli described above; orientation content centered on 45° or 135°; at 50% Michelson contrast) were presented at 2 Hz during a block (2 s cue, 14 s stimulus presentation). Participants were informed at the start of each stimulus presentation block with a cue (2 s) whether to either attend toward the stimuli, or to attend away from the grating (Supplementary Fig. 6). During attended stimulus blocks, observers performed an

orientation discrimination task, detecting and discriminating a change in the orientation of the stimulus compared with the global orientation (45° or 135°), target stimuli appeared with a probability of 60% throughout the stimulus block. To match task difficulty for the orientation task across observers, we titrated individual thresholds to yield an accuracy of 75%. During unattended stimulus blocks observers performed the same fixation task as described above; target letters appeared with a probability of 30%. All stimulus presentation blocks were completely identical, as both orientation and target letters would appear throughout a block, and only the initial cue informed the participant which task to perform (Supplementary Fig. 6). The two attentional condition blocks were pseudo-randomized over the course of the run, each interleaved with 16 s baseline periods. Behavioral data indicated performance for both tasks was well above chance (attended task: mean = 0.75, S.E.M. = 0.05, unattended task: mean = 0.88, S.E.M. = 0.04). Participants completed 5–10 fMRI runs; each run took 272 s to complete (8 stimulus blocks per run).

**Spatial attention experiment.** To directly assess whether attention modulates local gain control, we manipulated covert spatial attention for both stimuli configurations. Participants (n = 6) were instructed to maintain their gaze within a fixation circle (diameter, 1°) at the center of the display. Observers viewed stimuli that were the linear combinations of the same bandpass-filtered noise stimuli described above (outer diameter 15°; inner diameter 3°; spared midline), resulting in either orthogonal (45°/135° and 135°/45°) or collinear (45°/45° and 135°/135°) stimuli. Each individual component was rendered at 25% Michelson contrast, resulting in a combined grating of 50% Michelson contrast. Note that the overall contrast of these superimposed stimuli was lower than in the previous normalization experiment, as we wanted to leave enough headroom in the response for the attentional manipulation to take effect.

A cue (2 s) at the start of each block informed the participant to allocate their covert spatial attention to either the left or right side of a central fixation point, and remained displayed throughout the block (16 s total block duration; Supplementary Fig. 9). Observers performed a demanding probe detection task, detecting and discriminating whether a neutral gray Gaussian disk appeared at a random location within the upper or lower visual field on the attended side of fixation (probe size 1.5°, with smoothed edges). Probes could appear on either side of fixation throughout a stimulus block. However, observers were instructed to only respond to targets presented on the attended side, as indicated by the cue presented at fixation. The stimulus configuration displayed on either side of fixation was held constant within a block. Stimulus configuration and attention conditions were counter-balanced and presented in a pseudorandom order, and were interleaved with fixation blocks of equal duration. The behavioral ability to discriminate between targets was comparable for both stimuli configurations, as confirmed by measures acquired outside the scanner (collinear: mean = 0.87, S.E.M. = 0.03; orthogonal: mean = 0.90, S.E.M. = 0.03; paired t-test: t(5) = 0.502, p = 0.637). Participants completed 8–14 fMRI runs; each run took 272 s to complete (8 stimulus blocks per run). In addition, a scan session included two visual localizer scans, in which a flickering and rotating contrast pattern was presented within the same aperture as the stimuli (blocked presentation, 16 s on and off; 6 stimulus blocks per run).

**fMRI data acquisition and preprocessing.** MRI data were acquired at Harvard University's Center for Brain Science Neuroimaging Center (Cambridge, Massachusetts). Data for the first two experiments were collected in a single scan session, using a 3.0 Tesla Tim Trio MRI Scanner (Siemens, Erlangen, Germany) equipped with a 32-channel head coil. A scan lasted 2 h, during which we acquired: an anatomical scan (voxel size: 1.2 mm isotropic) using a T1-weighted multi-echo MPRAGE sequence, and functional volumes with whole brain coverage using a simultaneous multislice (SMS) acquisition protocol (69 slices, TR = 2 s, TE = 30 ms, flip angle = 80°, FoV = 216 mm, voxel size = 2 mm isotropic, in-plane acceleration factor 3, multiband factor 3[75,76]. The final experiment was collected using a 3.0 Tesla Prisma MRI Scanner equipped with a 64-channel head coil. A scan lasted 1.5–2 h, during which we acquired: an anatomical scan (voxel size: 1.2 mm isotropic) using a T1-weighted multi-echo MPRAGE sequence, and functional volumes with whole brain coverage using a SMS acquisition protocol (72 slices, TR = 2 s, TE = 30 ms, flip angle = 80°, FoV = 208 mm, voxel size = 2 mm isotropic, in-plane acceleration factor 3, multiband factor 3[75,77,78]. All analyses were performed in the native space for each participant. Functional volumes were aligned to reconstructed anatomical data, using a surface-based registration between the structural and functional MRI volumes implemented in Freesurfer[79]. Functional data were preprocessed using standard motion-correction procedures, Siemens slice timing correction, and boundary-based registration[79,80]. To optimize voxel-wise analyses, no volumetric spatial smoothing was performed. Robust rigid registration[81] was performed to align experimental data within each scan session, using the middle time-point of each scan. All further analyses were conducted using custom code written in Matlab.

**Regions of interest.** Population receptive field data collected during a separate scan session were analyzed using the 'analyzePRF' Matlab toolbox, and used to define regions of interest up to area V3[54,55]. Not all subjects were available for pRF

scanning; for one participant in Experiments 1 and 2, and one participant in Experiment 3, we defined retinotopic regions based on traditional retinotopy scans following standard procedures[82]. Within the regions of interest, we defined the top 25% of voxels based on the independent localizer scans (using a standard GLM analysis, selecting the most visually responsive voxels by their respective significance values) for those voxels whose estimated population receptive field (pRF) location fell within the stimulus aperture (15° diameter). For the participants that did not complete pRF scanning, a fixed localizer significance cut-off was used that yielded a similar number of total voxels, compared with the other participants. This voxel selection ensured that our analysis would be based on voxels that were visually responsive to the localizer stimulus.

**fMRI data analysis.** The preprocessed and aligned raw MRI time series per scan, for each voxel, was detrended, high-pass filtered and converted to percent signal change. Task data for all experiments were analyzed by obtaining the activity pattern for each stimulus block, and temporally averaging the BOLD activity across all block of the same condition for every voxel within the ROI, after time shifting by three TRs to account for the hemodynamic lag (Supplementary Figs. 1, 6 & 9). For Experiment 1, we quantified the difference between the BOLD response evoked by the orthogonal or collinear stimulus configurations by computing a difference, where a positive difference signals stronger normalization (Fig. 3a). For Experiment 2, we similarly quantified attentional modulation as the difference between attended vs. unattended blocks (Fig. 3b).

To compare the degree of voxel-wise dependency between these two measures we computed a Spearman correlation, for the voxels that passed the stringent voxel selection described above within the defined V1–V3 regions, which was Fisher-Z transformed to allow for comparison between observers (Fig. 4a). To ensure that the correlations were not driven by a signal-to-noise ratio (SNR) difference, our voxel selection was broken up into four equal bins, demonstrating similar correlations within each bin (Fig. 4b). To ensure that outliers did not drive the computed correlations, we discounted voxels for this voxel-wise correlation analysis that exceeded the mean normalization strength or attentional modulation measure (for each observer) by more than 3 s.d. (between 0 and 8 voxels were discounted for each observer, within a respective region).

For Experiment 3, covert spatial attention was manipulated to either the left or right side from central fixation, leaving the opposite visual field unattended. This allowed us to examine the effect of attention when attention was either directed toward or away from either visual field, for both collinear and orthogonal stimuli configurations (Supplementary Fig. 9). To quantify the magnitude of attentional modulation, we computed the difference between attending toward vs. attending away (Fig. 5).

**Modeling image statistics.** To assess the image statistics of our two stimuli configurations we first analyzed the power of the two image classes in the frequency domain using a standard 2-D Fourier transform. We generated 1000 unique bandpass-filtered noise images, which were combined either in a collinear or orthogonal configuration, resulting in 500 overlaid stimuli within each image class (see the Apparatus and Stimuli section; the size was matched to the screen resolution and visual angle to those images used in the experiments, so that these images were identical to the ones participants viewed in the scanner). Collapsing the Fourier domain power at each frequency band, over all orientations, confirmed that both image types carry the most power (beside the DC component) in those frequency bands the bandpass filter spared (2–3 cycles/°, see above; Fig. 2a).

Next, we constructed a V1-based energy detection model to describe a plausible underlying neural mechanism that resolves the discrepancy between the image statistics and the evoked BOLD response for each image configuration. We fed the same set of 1000 images used for the Fourier analysis into the V1-based energy detection model consisting of a bank of linear filters[44–49]. Because edge effects can introduce spurious output, images were not masked to match the circular stimulus configuration, and they were padded on the sides with 20% of the stimulus size (resulting in an image resolution of 775 × 775 pixels). The bank of linear filters consisted of 36 spatial frequencies (evenly spaced between 0.5 and 4 cycles/°), 8 orientations (evenly spaced between 0 and 180°), 2 phases (0, pi/2), with a receptive field size of 2° visual angle (Fig. 2b). After convolving each pixel within the image with following filters, we combined the quadrature-phase pairs analogous to a complex-cell energy model[45,83]:

$$\mathrm{Energy}_{\mathrm{pos,or,sf}} = \sqrt{\sum_{\mathrm{ph}} \left( \mathrm{image} \cdot \mathrm{filter}_{\mathrm{pos,or,sf,ph}} \right)^2} \qquad (1)$$

where $\mathrm{Energy}_{\mathrm{pos,or,sf}}$ represents the complex-cell energy for each image at a given position, orientation and spatial frequency, and $\mathrm{filter}_{\mathrm{pos,or,sf,ph}}$ represents the Gabor filter at each particular position, spatial frequency, and phase.

The contrast energy for all quadrature pairs were summed across spatial frequency scales and averaged over space, resulting in a measure of pooled contrast energy within each orientation channel (Fig. 2b):

$$\mathrm{TE}_{\mathrm{or}} = \frac{\sum_{\mathrm{pos}} \left( \sum_{\mathrm{sf}} \mathrm{Energy}_{\mathrm{pos,or,sf}} \right)}{\mathrm{num\_pos}} \qquad (2)$$

where $TE_{or}$ represents the total stimulus contrast energy at a given orientation, and num_pos reflects the total number of pixels within an image ($775 \times 775$).

Each images' summarized complex-cell output undergoes untuned divisive normalization, effectively acting as a contrast gain control operator[2,6,50]:

$$R_{untuned} = \sum_{or} \frac{TE_{or}{}^n}{\left(\left(\frac{\sum TE_{or}}{num\_or}\right)^n + \sigma\right)} \quad (3)$$

where, $R_{untuned}$ represents the normalized stimulus energy of each image, num_or reflects the total number of orientation channels, $\sigma$ is a constant (constrained at 0.5), and $n$ reflects the nonlinearity in the gain of the response (constrained at 1). For illustrative purposes we computed the total mean stimulus energy over all 1000 images (combining both collinear and orthogonal image configurations) to demean each output, and the maximum stimulus energy over all images to normalize to 1 (Fig. 2c).

Here, we incorporate a tuned component to the normalization pool[84], to account for the stronger sub-additive response to the collinear stimulus configuration. The untuned component pools equally over all oriented filters (see Eq. (3)), while the tuned component only contains information in the orientation channel matching the orientation image statistics:

$$R_{tuned} = \sum_{or} \frac{TE_{or}{}^n}{\left((\omega_{or} TE_{or})^n + \left(\frac{\sum TE_{or}}{num\_or}\right)^n + \sigma\right)} \quad (4)$$

where, $R_{tuned}$ represents the normalization stimulus energy of each image, $\omega_{or}$ reflects an array of the same size as $TE_{or}$, and only the orientation channel matching the collinear stimulus configuration statistics are non-zero, allowing a contribution of tuned normalization (i.e., the tuned component could only modulate energy in the 45° orientation channel; Fig. 2d). Future research measuring the full neural contrast response function, and manipulating features such as the size and shape of the attentional window, will shed more light on precisely how this contribution of feature-tuning is best incorporated into the normalization model of attention.

**Eye-position monitoring**. Eye-tracking data were acquired for 5 out of 6 observers for Experiments 1 and 2 (collected within the same scan session), and for 6 out of 7 observers for Experiment 3, using a MR-compatible SR Research EyeLink 1000 system (sampled at 1 kHz). After removing blinks, the mean distance from fixation was computed during time windows corresponding to the stimuli blocks. Specifically, we first calculated the x- and y-deviations, and then for Experiments 1 and 2 computed the absolute distance from fixation, while for Experiment 3 we focused on the x-trace displacement as it gives a better indication whether participants made eye movements toward the cued attended side. In both scan sessions eye movements were not >0.25° from central fixation (first scan session: 0.24°; second scan session: 0.18°), remaining well within the fixation circle (diameter 1°). However, one observer in Experiment 3 made eye movements >1° toward the attended side and was excluded from further data analysis. Importantly, for all other observers eye movements did not differ between stimulus configurations, when cued to attend either side of the visual field (repeated measures ANOVA: $F(1,5) = 0.92$, $p = 0.381$).

**Reporting summary**. Further information on research design is available in the Nature Research Reporting Summary linked to this article.

## Data availability

We have uploaded all preprocessed fMRI and behavioral data, to the Open Science Framework (OSF) at https://osf.io/4qz37. A reporting summary for this Article is available as a Supplementary Information file.

## Code availability

The experimental scripts used during data collection and analysis code to generate all figures in the manuscript are available from the Open Science Framework (OSF) at https://osf.io/4qz37.

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

## Acknowledgements

We thank Louis Vinke & Sara Aghajari for assistance with data collection, Janneke Jehee, Joe McGuire, Rosanne Rademaker and members of the Ling Lab for helpful comments and suggestions. We thank Himanshu Bhat and Thomas Benner (Siemens Healthcare) and Steven Cauley (MGH) for modifying and supplying the simultaneous multislice-BOLD imaging sequence. This work was supported by NIH EY028163 (S.L.).

## Author contributions

I.M.B. and S.L. conceived and designed the experiments, I.M.B. collected the data. I.M.B. conducted data analyses, with assistance from S.L. I.M.B. and S.L. wrote the paper.

## Competing interests

The authors declare no competing interests.
