## [Peer Review File · Nature Communications]

Reviewers' Comments:

Reviewer #1:

Remarks to the Author:

The manuscript provides functional imaging data from early visual cortex in humans that shows that non-linear summation effects and modulation with attention can be described by a tuned-normalization model. Normalization models in which the response of one neural element is divisively suppressed by the summed activity of a larger pool of neural elements have been used extensively to explain non-linear summation in visual cortex as well as a variety of effects on stimulus response by attention. The novel aspect of this study is to provide evidence in humans that the normalization pool is tuned to the same preferred orientation. Furthermore, that effects of directing spatial attention to the stimuli are larger when the tuned normalization is larger. These effects are summarized by a normalization model that has a tuned normalization component and where attention works to decrease only the suppressive component of response. Providing a formal model of attentions effects in visual cortex particularly by extending the normalization approach is quite a welcome approach to understanding attentional effects in humans and should influence others to adopt a more quantitative and mechanistic approach to understanding attention in humans. However, as presented, I have serious concerns including about how the model is fit and the rationale for how attentions effects are accounted for.

1) Tuned normalization fit. The logic proposed by the authors to explain the smaller sublinearity in the orthogonal case is that the tuned normalization component causes weaker suppression because the stimulus energy is now spread over two orientation components and therefore the tuned suppression is half that of the collinear case. This seems sensible. However, the model does not appear to capture this logic. Rather than just fit one weight (ω) to the full data set and allow this effect to cause larger or smaller responses for orthogonal and collinear responses, there are apparently two weights on the tuned normalization component, one for orthogonal stimuli and one for collinear stimuli (Figure 6a, incidentally, the stimulus drive pictures appear to be mixed up - the top one shows a single orientation stimulus drive, for the stimulus with two orientations?). This would suggest that the gain on tuned normalization is changed between stimulus conditions which seems unrealistic; i.e. what would cause this change in gain between conditions? Moreover, there are only two conditions - collinear and orthogonal - you could fit two weights in almost any part of the equation to get two different responses? Instead, following the logic provided, there should just be one weight that controls how tuned normalization is and then let this effect cause the differences in sublinearity.

2) Evaluating model fit. Before examining model parameters (6b-d), the model fit should be evaluated. While there are R^2 values in the text, this doesn't really show how well the model can quantitatively reproduce the differences between collinear / orthogonal responses and the attention effects. A great bulk of the R^2 value presumably comes from just being able to account for there being some response (i.e. right-hand side of 6a, also - does the HRF used have a scale parameter to convert to units of % signal change?). It would be helpful to show the model fits to all the conditions (i.e. supplementary figures 2 and 3, panel b), ideally on cross-validated data. Statistical tests (either nested F-test or AIC/BIC or some other statistics that accounts for number of parameters) should be shown to evaluate whether the ω weight and attentional gain, γ , actually account for a significant amount of variance relative to models without these terms.

3) Attentional gain only on normalization. In equation 5, attentional gain is only applied to the normalization component (as a divisive decrease in suppression). This does not seem justified. The Reynolds & Heeger formalization has attentional gain occurring on both the stimulus drive and thus propagating into the suppressive drive. What evidence is there that attentional gain only works on the normalization component and not the stimulus drive? Moreover, that it acts only as a decrease in gain

of this component?

4) Sub-additivity index or raw response difference. In the section that analyzes passive viewing of orthogonal or collinear oriented stimuli the index used to quantify sub-additivity should be clearly specified and used to assess the phenomenon. In particular, logically it would seem like the best way to assess sub-additivity is to report how much the combined stimulus response is relative to the sum of the individual components. This would mean measuring each individual component 45 and 135 separately, then summing two 45 (or 135) responses together for the collinear 45 (or 135) case and the 45 and 135 responses for the orthogonal case. Then the ratio of the actual response to the compound stimuli to these predictions reported. However, it appears that only one stimulus component was measured and that response was doubled. This could be a potential problem if the responses to the two orientations are not equal in magnitude (particularly possible on a voxel-by-voxel basis). Moreover, it is not made clear by the authors what is being reported in the text and figure, what is plotted in Figure 1c for instance? An index? Raw difference?

5) Adaptation. One possible alternate explanation for the difference in collinear and orthogonal responses is that there is stronger adaptation to putting 100% contrast into one orientation band compared to 50% contrast in separate orientation bands. Across 14 seconds of stimulus presentation adaptation could occur and be stronger for the collinear case and result in a smaller response compared to the orthogonal condition. This possibility should be considered and discussed.

6) Gabor filter model. How were edges of stimulus treated? There is some literature on aperture effects on oriented stimuli, is the model evaluated at the edges of the stimuli? In the text it says the stimulus energy was demeaned and normalized to 1. Does normalization here refer to z-scoring (i.e. normalizing the standard deviation)? For Figure 2 and the description, some reference to the methods would be helpful.

7) Untuned normalization strength and attention. The authors concentrate on how differences in tuned normalization correlate with attentional effects. Is there also a correlation with untuned normalization and attentional gain? Wouldn't the model which uses gamma to scale the whole normalization component predict this?

8) Hinges. The framing of attention as hinging on tuned normalization strength seems vague. What does hinging mean in this context? That you have to have tuned normalization for attention effects? That is not what is shown - particularly because the model would have attention working on both the tuned and untuned components of normalization. I would find the paper much easier to follow if it started with the normalization model of attention and then showed why it needs to be extended in human data to have a tuned component and the framing simply as support for a tuned normalization model of attention.

Reviewer #2:

Remarks to the Author:

Bloem & Ling sought to evaluate whether changes in the extent to which a stimulus recruits a normalization pool impacts the enhancement in neural responses induced by attention. They reasoned that if one role of attention is to reduce the suppressive influence of the normalization pool (as described in Reynolds & Heeger, 2009, cited by the authors), stimuli that are more strongly normalized (suppressed) are those which are subject to the greatest enhancement (via release from that suppression). They designed a really interesting and effective set of experiments to characterize (1) whether normalization exhibits feature tuning as measured with fMRI, (2) whether voxels that

show the greatest amount of normalization (assayed with subadditivity based on linear assumptions) are those with the greatest attentional modulations, (3) whether changing the amount of tuned normalization in a stimulus changed the amount of attentional enhancement observed, and (4) whether an adaptation of the Reynolds & Heeger modeling framework could account for these results. They found a strong component of tuned normalization in visual cortical responses measured with fMRI that correlated with the degree to which a voxel is enhanced by attention, consistent with their predictions. Then, they combined these findings to make predictions for the final experiment, which showed that manipulating the stimulus to induce greater normalization (and thus greater suppression) led to greater attentional enhancement. Finally, they claimed that a simplified normalization model invoking tuned normalization could account for their results.

Overall, I think this report communicates a really nice set of experimental results, and would be of broad interest to the readership of Nature Communications. I also believe the authors have demonstrated, quite clearly, something important that has not, to my knowledge, been shown before in humans: a population-level measure derived from sensory stimulation can predict attentional modulation in a straightforward manner. This alone would make the manuscript a strong candidate for this journal, and I think is the strongest part of the manuscript. The final experiment (Fig. 5) is icing on the cake (but also begs some important analysis questions, which I address below). However, I'm less enthusiastic about the modeling approach, and think the statistical treatment of the results could be improved. Regarding the model, I think the authors will need to either clearly describe why the current form of the model adds something meaningful to the paper, and/or generate a set of testable predicted results that can be evaluated in future studies. I anticipate the authors can address these issues with edits to the manuscript and/or updates to their analyses. I list major and minor concerns below:

Major:

1. While I find the experimental approach to be principled, I'm not sure I understand the motivation behind relating tuned normalization to attentional modulations. In the introduction, the authors first discuss the ubiquity of normalization in the visual system (as well as the known importance of tuned normalization), then introduce attention as inducing gain on neural responses, and transition to discussing the use of normalization in models of attention, the most prominent example of which is the Reynolds & Heeger 2009 model. Next, they discuss the possibility of including feature-tuned normalization in these types of models. But then, in the next paragraph, they make a leap to their hypothesis "that attention-driven modulation of the gain of responses within human visual cortex is made possible, in part, by a release from feature-tuned normalization". In the original formulation of the Reynolds & Heeger (2009) normalization model of attention (NMA), attention acts to increase stimulus-evoked responses, which are convolved with a suppressive field before normalizing those attention-enhanced responses – that is, attention acts at both the numerator and the denominator. This joint role of attention is critical for accounting for the diverse range of changes in contrast response functions and feature tuning functions observed in the primate literature (and, subsequently, the human behavioral and electrophysiology literature). But, nowhere in the original formulation of the NMA is there the notion that attention acts to reduce normalization, or to act in a way that's dependent on the extent of suppression (within a fixed attention field), at least not that I can find. That's not to say I don't think the authors' hypothesis is quite interesting (and their data very compelling) – I do, and it is! – it just needs a bit more build-up and rationale, and it should be made a bit more clear that this is a bit outside the 'standard' formulation of the Reynolds & Heeger NMA (which is cited in the Methods when describing the modeling procedures). Of course, if I misunderstood the rationale for or implementation of the model, I am happy to be corrected by the authors, but I anticipate similar misunderstandings are possible for other readers in the manuscript's present form.

2. Related to #1, the relationship of the authors' proposed model to the canonical NMA should be more clearly spelled out, especially the lack of a multiplicative attention field in its current form (Fig. 6, Eq. 5). I'll note that the current formulation effectively involves a multiplicative gain in the

numerator (since multiplying by λ^{-1} in the denominator is the same as multiplying by λ in the numerator). When rewritten this way, this current formulation is quite similar to the model proposed by Boynton (Boynton, *Curr Op Neurobio*, 2005): that model used a gain field multiplied by a stimulus drive and normalization factor, but allowed the gain field to be negative to account for suppressive effects of attention. But, in the authors' model here, there is no role of attention in the normalization pool (denominator), which is a significant and important departure from the structure of the original NMA. Again, I think this model is certainly interesting and worth presenting/exploring, but its relationship and divergence from the 'classic' model is necessary to point out. The authors may also wish to relate their model to that proposed in the Ni et al, 2012 report they cite. While that one accounts for single-cell responses, it could still be interesting to discuss.

3. At present, when fitting the model shown in Fig. 6 and Eq. 5, it's not clear why fitting was performed separately for each stimulus configuration. Shouldn't a voxel have a single measure of tuned normalization that applies to all stimulus configuration conditions? That is – in each configuration, the voxel should have the same static encoding model describing the degree to which its responses are normalized by different feature values (in this case, just 2 possibilities). My sense is that this model should be constrained by measured responses across both configurations. Ideally, the full span of relative orientations would be tested here to measure each voxel's (feature) normalization kernel, but even with the 2-level experiment here, the same logic should apply I think. In any case, if the authors have a strong argument for why it's appropriate to fit the encoding model separately for each stimulus configuration, this should be more clearly described in the text.

4. Related, isn't the result shown in Fig. 6b and the left panel of Fig. 6c necessarily true given the modeling framework (Eq. 4 for tuned normalization component only)? The only free parameter in the model is w_{or} , which is allowed to vary across conditions. When the measured response is smaller (in collinear case), w_{or} must increase, since all other terms stay the same (based on arguments presented in Fig. 2). Am I missing something here? See the next comment, but perhaps there's a way to combine my comment 3 above, and 5 below, to fit encoding models for each individual voxel across stimulus configurations, then use those to predict attentional modulation on the single-voxel level. Otherwise, this modeling procedure seems to just reveal that you need to increase the denominator to fit a smaller number and multiply by a constant in the numerator to fit attentional gain. Related, the last sentence in the first paragraph on pg 13 ("Finally, we propose...") does not seem justified given the current model-fitting procedure: what aspects of the results were predicted, rather than just fit condition-by-condition? Fig. 4 is the closest, but there's no model there – just a comparison of measurements. As mentioned above, I think the manuscript could benefit from a set of testable predictions borne out by the model proposed by the authors (even if they are not tested in the present study).

5. Is there a way to link Figs 4 & 5? Fig 4 shows that, across tasks, the degree of tuned normalization estimated when attention is directed away is correlated with the amount of attentional modulation in a voxel when attention is directed towards a stimulus. Fig 5 shows the results of the experiment in which both tuned normalization and attention were manipulated together. Can a similar voxel-by-voxel correlation to that shown in Fig. 4 be shown using data from Expt 3? Trials in which the contralateral stimulus was unattended could be used to estimate the amount of normalization in each voxel, similar to Fig. 4, and then the impact of attention on that voxel's response could be compared for both configurations (correlate attend-ortho with the tuned normalization index for unattended, then attend-collinear with the same tuned normalization index). It may even be possible to compare, voxel-by-voxel, the degree to which the attentional enhancement depends on stimulus configuration: AMI-ortho vs AMI-collinear. In any case, I think some exploration of similar correlations to those shown in Fig. 4 would make the authors' case even stronger.

6. I'm not sure I'm convinced the authors have demonstrated that attention acts to reduce suppression induced by tuned normalization. As described above (Comment 2), the current model of attention acts on both the tuned and untuned normalization (and really, just acts as a general gain control in the numerator). To demonstrate that attention acts to reduce the impact of tuned

normalization, the authors would need to demonstrate that attention primarily impacts the tuned normalization component of the model – that is, they'd need to move the $\lambda-1$ in front of the tuned component of Eq 5 (rather than distributing it across all normalization terms in the denominator). I'm not sure the present dataset would afford this test (not enough unique conditions tested). But, again, I may be missing something – I'm happy for the authors to correct me and clarify the manuscript as necessary to avoid reader misunderstanding.

7. Throughout the manuscript, the statistical comparisons could be improved. At present, there seems to be no correction for multiple comparisons when appropriate (e.g., across ROIs), and the authors use different metrics across experiments without justification (see Minor Point 6 below). There are also a few results with p's hovering above 0.05 that are listed in support of a result (e.g., Fig. 5c, pg 10, V3). There are also cases where ANOVAs could be used to test for main effects/interaction, such as when comparing effect of stimulus configuration across ROIs.

Minor:

1. Fig. 1c: I'm not sure I understand the x/y axis labels. Is each label "orthogonal/collinear minus additivity"? If so, shouldn't most voxels be negative? It seems the point is to show that, voxel-by-voxel, response are more subadditive for the collinear configuration than the orthogonal configuration. Additionally, the description of this result at the end of the paragraph at the top of the next page suggests that whatever is labeled collinear should be lower than whatever is labeled orthogonal (consistent with Fig. 1a), yet this is not the immediate visual impression one gets from looking at Fig. 1c. Additionally, in keeping with the exceptionally transparent data presentation of Fig. 4, it would be useful to show this for the 3 ROIs

2. Pg 6, the authors say "the magnitude of the feature-tuned aspect of normalization seemed to decrease in strength along the visual hierarchy", but do not provide any statistical basis for this claim – if it's just a qualitative statement, specifying this would help (though if it could be statistically supported, that would be even better)

3. Have the authors considered evaluating the reliability of the tuned normalization measure they compute per voxel? Perhaps something like split-half reliability, etc, could be useful to know (especially if this measure is meant to be used as a potential proxy for impact of attention on neural responses)

4. I'm also curious if the authors have evaluated the extent to which tuned normalization might vary across the visual field (and thus across the cortical surface) – is tuned normalization greater near the edges of the stimulus? The authors mention in the methods that they measure single-voxel receptive fields for most subjects, so it may be possible to examine this in the present dataset. Even looking at the amount of tuned normalization projected onto the cortical surface (and next to retinotopic mapping data) could provide hints for what types of mechanisms could be driving these effects and inspire future studies.

5. As a point of clarification – in Experiment 3 are both stimuli always presented with the same configuration on each trial?

6. Figs 3&4 and Fig. 5 use different metrics of signal modulation: Figs 3&4 (Experiments 1 & 2) compute a simple difference measure, while Fig. 5 (Experiment 3) uses a commonly-used attentional modulation index measure (difference/sum). Either metric is fine, but it would be easiest to stick to one throughout the manuscript (or justify why different metrics are appropriate). On pg 22, they state that an AMI is "suggested to be a better representation when comparing an attentional effect between different conditions, as it is not biased by the differences in BOLD responses evoked by the two different stimuli configurations", but provide no reference supporting this assertion. This may be helpful to include. If the authors decide to go with the AMI, it seems fair to also use a similar modulation index for the normalization metric. Note, of course, that modulation index measures are only useful for strictly-positive signals (like firing rates).

7. I overall really like Fig. 4. However, plotting data from the green subject up front and the blue subject in the back seems to highlight the strongest effect and hide the weakest one (based on the

correlation values plotted in Fig. 4b). Could each participant be plotted individually in a supplementary figure? Or is there a way to convey this without essentially masking all but one subject?

8. Fig. S3: it seems that the x-axis of the bar graphs in panel b is mislabeled? Those seem to be conditions, while the dots are observers.

9. For the model-fitting described on pg 11, was the model fit to each voxel, then best-fit parameters averaged across voxels, or was the model fit to the average response across all voxels in an ROI?

10. Fig. 6a: I think the "stimulus drive" panels are flipped: the one to the right of "orthogonal" shows only a single orientation, while the one to the right of "collinear" shows two.

11. On pg 20 in the Methods, the authors disclose that behavioral performance was different between the attend-stimulus and attend-fixation tasks. Is this difference significant? Could behavioral performance differences account for any aspect of the results?

12. For Experiment 3, what were the 'probe' stimuli? Neutral gray discs? And what was behavioral performance inside the scanner?

13. The authors should include a data/code availability statement for the fMRI data and the computational models. I recommend posting data and code on a publicly-accessible repository, such as Open Science Framework and/or GitHub.

Reviewer #3:

Remarks to the Author:

General comments:

The authors employed fMRI and computational modeling to explore the relationships between feature-tuned normalization and attentional modulation of visual cortical responses. They report that the fMRI responses evoked by an overlaid pair of oriented stimuli are lower than the additive sum of the individual oriented stimuli and that this sub-additivity of responses is more pronounced for collinear compared to orthogonal stimuli. The authors then correlated this measure of feature-tuned normalization with modulation of responses based on whether subjects directed attention towards or away from the stimulus. They report a correlation of these two measures across voxels and participants. An additional experiment in which different parts of the stimulus were either attended or unattended provides further evidence for a relationship between feature-tuned normalization and attentional modulation. Finally, an extension of the normalization model of attention that includes feature-tuned normalization is described and shown to account for the empirical results.

This is an elegant and well-designed study that provides important new information about the mechanisms of attentional modulation, feature-based normalization, and their interaction. The correlation analysis shown in Figure 4 currently conflates between- and within-subject variation in attention modulation and tuned normalization strength, and this should be rectified. Also, adding results from a model comparison would strengthen the modeling findings. Minor issues and suggestions are listed below.

Specific comments:

Major points:

1) The correlation analyses presented in figure 4 are a combination of correlations due to individual differences in attention modulation and tuned normalization strength and those due to correlations of these measures across voxels in each subject. However, the authors' interpretation is that the correlations are due to variability across voxels in each subject. The authors should recalculate the correlations after mean-centering the data for each participant.

2) It is difficult to evaluate the importance of the modeling results, as only one model is described. The authors should conduct a quantitative model comparison with one or more alternative models that

do not contain feature-based normalization (taking into account differences in the numbers of free parameters in the models).

Minor points:

1) Figures 1, 3, 4, and 5 present data from individual subjects in which each subject is assigned a color along a gradient (green to blue or dark gray to light gray). In my opinion, this data visualization is ineffective, as the colors representing the individual subjects are not distinguishable enough for the reader to be able to assess individual differences. Plotting the individual data points for each subject is commendable, but I think it would be best to use the same color for all participants.

2) The results of statistical tests comparing BOLD responses to orthogonal and collinear stimuli are presented twice, once on page 5 and again on page 7.

3) page 9, "The less pronounced relationship in V3 is likely driven by the reduced heterogeneity and overall magnitude of tuned normalization strength observed in this visual area." Are the strength of correlation and the heterogeneity and magnitude of tuned normalization strength actually lower in V3 compared to V2?

4) page 8, Figure 4 legend, "A tight relationship between tuned normalization strength and attentional modulation is evident for the top 25% selected voxels for each observer." At this stage in the paper, it is unclear what is meant by "top 25% selected voxels", and this should be more clearly explained here.

5) page 11, "Results revealed that the contribution of tuned normalization in the absence of attention for collinear configurations was larger..." I think it would be more accurate to write something like "Results revealed that when attention was directed away from the stimulus, the contribution of tuned normalization was larger for collinear configurations..."

6) The pair of orientation/space plots in Figure 6 that represent the outputs of the V1-energy model appear to be reversed. Presumably the one that has energy at two orientations should be at the top (corresponding to the orthogonal configuration), and the one with energy at a single orientation should be at the bottom (corresponding to the collinear configuration).

7) page 13, "While the notion of feature-based attention is well established(59,60), incorporating this into the normalization model does not account for the results we have presented here." What is the evidence for this claim?

8) page 14, "While neuroimaging studies using a typical center-surround stimuli often report an attenuation of the response to the center stimulus..." One of the references listed here (Xing and Heeger, 2000) is not a neuroimaging study.

9) page 14, "We hypothesized that by keeping the orientation of one of the components constant and manipulating the orientation of the second component, we can induce normalization more analogous to surround suppression within all neural populations with receptive fields falling within our stimulus bounds. The superimposed configurations indeed elicited the predicted populations responses that one would expect from tuned normalization, as we found lower BOLD responses for those configurations that matched in their orientation content, compared to configurations with orthogonal orientation information." Some references to the literature of orientation selectivity of surround suppression (both at the perceptual and neural level) would be helpful here.

10) page 19, "A power analysis indicated that six participants would be sufficient to detect the

reported normalization strength and attention effects." More information should be provided about this power analysis. Did it incorporate variances and effect sizes taken from previous studies of normalization strength and attention effects?

11) I don't think that the properties of the probe used in the spatial attention experiment are described anywhere in the paper.

12) page 21, "Within the regions of interest, we defined the top 25% of voxels based on the independent localizer scans (using a standard GLM analysis) for those voxels whose estimated population receptive field (pRF) location fell within the stimulus aperture (15 degrees diameter)." How were the top 25% of voxels defined for those subjects who did not complete pRF scanning and instead participated in retinotopic mapping with periodic stimuli?

13) page 22, "This attention index is suggested to be a better representation when comparing an attentional effect between different conditions,..." Better than what?

14) page 24, Equation #3. The value of exponent n , which quantifies nonlinearity in the gain of the response, was constrained to be 1. The authors should explain how this value was chosen and the significance of this choice for the conclusions they draw from the modeling.

Reviewer 1:

“The manuscript provides functional imaging data from early visual cortex in humans that shows that non-linear summation effects and modulation with attention can be described by a tuned-normalization model. Normalization models in which the response of one neural element is divisively suppressed by the summed activity of a larger pool of neural elements have been used extensively to explain non-linear summation in visual cortex as well as a variety of effects on stimulus response by attention. The novel aspect of this study is to provide evidence in humans that the normalization pool is tuned to the same preferred orientation. Furthermore, that effects of directing spatial attention to the stimuli are larger when the tuned normalization is larger. These effects are summarized by a normalization model that has a tuned normalization component and where attention works to decrease only the suppressive component of response. Providing a formal model of attentions effects in visual cortex particularly by extending the normalization approach is quite a welcome approach to understanding attentional effects in humans and should influence others to adopt a more quantitative and mechanistic approach to understanding attention in humans. However, as presented, I have serious concerns including about how the model is fit and the rationale for how attentions effects are accounted for.”

We thank the reviewer for their kind words and comments. As we have explained in our response to the Editor (see above), we have removed the tuned normalization model from this manuscript, which accounted for a number of concerns that were raised. Outlined below, we have addressed the remainder of Reviewer 1’s concerns.

1. *“The logic proposed by the authors to explain the smaller sublinearity in the orthogonal case is that the tuned normalization component causes weaker suppression because the stimulus energy is now spread over two orientation components and therefore the tuned suppression is half that of the collinear case. This seems sensible. However, the model does not appear to capture this logic. Rather than just fit one weight (ω) to the full data set and allow this effect to cause larger or smaller responses for orthogonal and collinear responses, there are apparently two weights on the tuned normalization component, one for orthogonal stimuli and one for collinear stimuli (Figure 6a, incidentally, the stimulus drive pictures appear to be mixed up - the top one shows a single orientation stimulus drive, for the stimulus with two orientations?). This would suggest that the gain on tuned normalization is changed between stimulus conditions which seems unrealistic; i.e. what would cause this change in gain between conditions? Moreover, there are only two conditions - collinear and orthogonal - you could fit two weights in almost any part of the equation to get two different responses? Instead, following the logic provided, there should just be one weight that controls how tuned normalization is and then let this effect cause the differences in sublinearity.”*

Although the formal tuned normalization model is no longer in this revised manuscript, this comment still applies to the simulation shown in Figure 2. We agree that incorporating two weight parameters for the tuned normalization component in our model was unnecessary, and have, as suggested, made the model more parsimonious by cutting it down to one normalization weight. In doing so, we now illustrate how the two stimuli energy distributions can be shifted apart by including a tuned component in the normalization pool for collinear stimuli. To reflect this change, the Methods section now states: *“Here, we incorporate a tuned component to the normalization pool²⁴, to account for the stronger sub-additive response to the collinear stimulus configuration. The untuned component pools equally over*

all oriented filters (see Equation 3), while the tuned component only contains information in the orientation channel matching the orientation image statistics:

$$R_{tuned} = \sum_{or} \frac{TE_{or}^n}{\left((\omega_{or} TE_{or})^n + \left(\frac{\sum TE_{or}}{num_{or}} \right)^n + \sigma \right)} \quad 4)$$

where, R_{tuned} represents the normalization stimulus energy of each image, ω_{or} reflects an array of the same size as TE_{or} , and only the orientation channel matching the collinear stimulus configuration statistics are non-zero, allowing a contribution of tuned normalization (i.e., the tuned component could only modulate energy in the 45° orientation channel; **Figure 2d**). Future research measuring the full contrast response function will shed more light on precisely how this contribution of the tuned component is incorporated in the normalization model.”, see page 23.

2. “In the section that analyzes passive viewing of orthogonal or collinear oriented stimuli the index used to quantify sub-additivity should be clearly specified and used to assess the phenomenon. In particular, logically it would seem like the best way to assess sub-additivity is to report how much the combined stimulus response is relative to the sum of the individual components. This would mean measuring each individual component 45 and 135 separately, then summing two 45 (or 135) responses together for the collinear 45 (or 135) case and the 45 and 135 responses for the orthogonal case. Then the ratio of the actual response to the compound stimuli to these predictions reported. However, it appears that only one stimulus component was measured and that response was doubled. This could be a potential problem if the responses to the two orientations are not equal in magnitude (particularly possible on a voxel-by-voxel basis). Moreover, it is not made clear by the authors what is being reported in the text and figure, what is plotted in Figure 1c for instance? An index? Raw difference?”

The responses evoked by the individual components were measured for both orientations (45° & 135°), and combined to create the hypothetical additive response. We did not find qualitative differences between these responses and therefore did not break up the sub-additivity measure as the reviewer suggested. The fact that the response to the 45° and 135° are similar makes the comparison (45°+45° or 135°+135° vs 45°+135°) equivalent. We now include results of these analyses in Figure S2, and we have clarified this in the manuscript, “Additionally, in a separate set of scans we measured the BOLD response to both individual components (45° & 135°), in separate blocks, that comprised the overlaid stimuli, and summed the responses for each component in order to create a hypothetical additive sum. Note, we did not find a difference in the evoked responses to individual components based on orientation (45° or 135°; **Figure S2**).”, see pages 5-6, as well as in the Methods, “Oriented stimuli (45° & 135°) were presented at 2Hz in separate blocks (2s cue, 14s stimulus presentation), and were interleaved with baseline periods.”, see page 18. Moreover, we agree that the axes labels of Figure 1c were ambiguous, and have clarified this in the manuscript as well as the figure legend, “Although there is heterogeneity in the magnitude of sub-additivity between voxels within a region, comparing the difference between the hypothetical additive sum with both collinear and orthogonal stimulus configurations revealed a consistent pattern, with the collinear configuration exhibiting larger sub-additivity compared to the orthogonal configuration –an effect that was highly reliable for almost all voxels within V1 (**Figure 1c, S3, & S4**).”, see page 6.

3. *“One possible alternate explanation for the difference in collinear and orthogonal responses is that there is stronger adaptation to putting 100% contrast into one orientation band compared to 50% contrast in separate orientation bands. Across 14 seconds of stimulus presentation adaptation could occur and be stronger for the collinear case and result in a smaller response compared to the orthogonal condition. This possibility should be considered and discussed.”*

This is an interesting point. To address it, we include additional analyses where we tested for the role of adaptation in explaining the differences between collinear and orthogonal responses. To do so, we fitted the evoked response (across the block, excluding the initial 2 TRs (4s) to account for the hemodynamic lag) in both stimulus configurations and compared slopes. If stronger adaptation occurred in Collinear configurations, we would expect the slope across this epoch to be negative, and steeper than the orthogonal condition. However, this analysis revealed no differences in slope between the two conditions, suggesting that adaptation did not play a major role in the difference between Collinear and Orthogonal conditions. We now touch on this possible interpretation in the manuscript: *“Importantly, the differences in BOLD responses evoked by the two stimuli configurations were not driven by differences in the image statistics, adaptation (see Figure S5), nor could they be explained by basic first-order visual response properties.”*, see page 6. Moreover, we include the results of these analyses in Supplementary Materials.

4. *“Gabor filter model. How were edges of stimulus treated? There is some literature on aperture effects on oriented stimuli, is the model evaluated at the edges of the stimuli? In the text it says the stimulus energy was demeaned and normalized to 1. Does normalization here refer to z-scoring (i.e. normalizing the standard deviation)? For Figure 2 and the description, some reference to the methods would be helpful.”*

The images that were passed through the model were not masked to match the stimulus configuration, also to avoid these edge problems. Furthermore, the whole image was padded on the sides, which avoided convolution-based edge effects. We now include more details about this in our Methods: *“Because edge effects can introduce spurious output, images were not masked to match the stimulus configuration, and they were padded on the sides with 20% of the stimulus size (resulting in an image resolution of 775x775 pixels).”*, see page 22. We have also added reference to those methods in the captions for Figure 2. We did not apply z-scoring to the stimulus energy; instead we demeaned overall stimulus energy of the whole population of images and normalized by its maximum value (collinear and orthogonal combined). This is mainly done for illustrative purposes, as the actual stimulus energy value is only relevant in proportion to either stimulus configuration.

5. *“Untuned normalization strength and attention. The authors concentrate on how differences in tuned normalization correlate with attentional effects. Is there also a correlation with untuned normalization and attentional gain? Wouldn't the model which uses gamma to scale the whole normalization component predict this?”*

While our experimental design was optimized to measure tuned normalization, a measure that would approximate the magnitude untuned normalization would be the deviation from sub-additivity for the orthogonal stimulus configuration. However, the sub-additivity measure is composed of the BOLD response to unattended condition measured in the second experiment. Computing the correlation between this deviation from sub-additivity

and attentional gain would therefore not be statistically independent. Thus, we could not appropriately examine the correlation between untuned normalization and attentional modulation.

6. *“The framing of attention as hinging on tuned normalization strength seems vague. What does hinging mean in this context? That you have to have tuned normalization for attention effects? That is not what is shown - particularly because the model would have attention working on both the tuned and untuned components of normalization. I would find the paper much easier to follow if it started with the normalization model of attention and then showed why it needs to be extended in human data to have a tuned component and the framing simply as support for a tuned normalization model of attention.”*

We agree the title was somewhat ambiguous, and commits too much to the tuned component of normalization, when it comes to attentional modulation. We have therefore changed the title to *“Normalization governs attentional modulation within human visual cortex”*. Within the manuscript, we also make it clearer now that tuned normalization is but a component of the normalization model, and it is likely that attention works on both untuned and tuned normalization. Furthermore, by removing the formal tuned normalization model from the manuscript, we no longer make a specific prediction about how attention and tuned normalization interact, acknowledging that further work is needed to tease apart the contribution of these components.

Reviewer 2:

“Bloem & Ling sought to evaluate whether changes in the extent to which a stimulus recruits a normalization pool impacts the enhancement in neural responses induced by attention. They reasoned that if one role of attention is to reduce the suppressive influence of the normalization pool (as described in Reynolds & Heeger, 2009, cited by the authors), stimuli that are more strongly normalized (suppressed) are those which are subject to the greatest enhancement (via release from that suppression). They designed a really interesting and effective set of experiments to characterize (1) whether normalization exhibits feature tuning as measured with fMRI, (2) whether voxels that show the greatest amount of normalization (assayed with subadditivity based on linear assumptions) are those with the greatest attentional modulations, (3) whether changing the amount of tuned normalization in a stimulus changed the amount of attentional enhancement observed, and (4) whether an adaptation of the Reynolds & Heeger modeling framework could account for these results. They found a strong component of tuned normalization in visual cortical responses measured with fMRI that correlated with the degree to which a voxel is enhanced by attention, consistent with their predictions. Then, they combined these findings to make predictions for the final experiment, which showed that manipulating the stimulus to induce greater normalization (and thus greater suppression) led to greater attentional enhancement. Finally, they claimed that a simplified normalization model invoking tuned normalization could account for their results. Overall, I think this report communicates a really nice set of experimental results, and would be of broad interest to the readership of Nature Communications. I also believe the authors have demonstrated, quite clearly, something important that has not, to my knowledge, been shown before in humans: a population-level measure derived from sensory stimulation can predict attentional modulation in a straightforward manner. This alone would make the manuscript a strong candidate for this journal, and I think is the strongest part of the manuscript. The final experiment (Fig. 5) is icing on the cake (but also begs some important analysis questions, which I address below). However, I’m less enthusiastic about the modeling approach, and think the statistical treatment of the results could be improved. Regarding the model, I think the authors will need to either clearly describe why the current form of the model adds something meaningful to the paper, and/or generate a set of testable predicted results that can be evaluated in future studies. I anticipate the authors can address these issues with edits to the manuscript and/or updates to their analyses”

We were pleased to see that Reviewer 2 found the experiments interesting, and “of broad interest to the readership of Nature Communications”. As mentioned above, we have removed the tuned normalization model from this manuscript, which accounted for a number of concerns that were raised. Outlined below, we have addressed the remainder of Reviewer 2’s concerns.

1. *“While I find the experimental approach to be principled, I’m not sure I understand the motivation behind relating tuned normalization to attentional modulations. In the introduction, the authors first discuss the ubiquity of normalization in the visual system (as well as the known importance of tuned normalization), then introduce attention as inducing gain on neural responses, and transition to discussing the use of normalization in models of attention, the most prominent example of which is the Reynolds & Heeger 2009 model. Next, they discuss the possibility of including feature-tuned normalization in these types of models. But then, in the next paragraph, they make a leap to their hypothesis “that attention-driven modulation of the gain of responses within human visual cortex is made possible, in part, by a release from*

feature-tuned normalization". In the original formulation of the Reynolds & Heeger (2009) normalization model of attention (NMA), attention acts to increase stimulus-evoked responses, which are convolved with a suppressive field before normalizing those attention-enhanced responses – that is, attention acts at both the numerator and the denominator. This joint role of attention is critical for accounting for the diverse range of changes in contrast response functions and feature tuning functions observed in the primate literature (and, subsequently, the human behavioral and electrophysiology literature). But, nowhere in the original formulation of the NMA is there the notion that attention acts to reduce normalization, or to act in a way that's dependent on the extent of suppression (within a fixed attention field), at least not that I can find. That's not to say I don't think the authors' hypothesis is quite interesting (and their data very compelling) – I do, and it is! – it just needs a bit more build-up and rationale, and it should be made a bit more clear that this is a bit outside the 'standard' formulation of the Reynolds & Heeger NMA (which is cited in the Methods when describing the modeling procedures). Of course, if I misunderstood the rationale for or implementation of the model, I am happy to be corrected by the authors, but I anticipate similar misunderstandings are possible for other readers in the manuscript's present form."

We agree with the reviewer that this was a big leap in reasoning and have changed our wording in the introduction, see pages 3-4. Furthermore, by removing the formal tuned normalization model from the manuscript, we no longer make a specific prediction about how attention and tuned normalization interact, acknowledging that further work is needed to tease apart the contribution of these components.

- 2. "Is there a way to link Figs 4 & 5? Fig 4 shows that, across tasks, the degree of tuned normalization estimated when attention is directed away is correlated with the amount of attentional modulation in a voxel when attention is directed towards a stimulus. Fig 5 shows the results of the experiment in which both tuned normalization and attention were manipulated together. Can a similar voxel-by-voxel correlation to that shown in Fig. 4 be shown using data from Expt 3? Trials in which the contralateral stimulus was unattended could be used to estimate the amount of normalization in each voxel, similar to Fig. 4, and then the impact of attention on that voxel's response could be compared for both configurations (correlate attend-ortho with the tuned normalization index for unattended, then attend-collinear with the same tuned normalization index). It may even be possible to compare, voxel-by-voxel, the degree to which the attentional enhancement depends on stimulus configuration: AMI-ortho vs AMI-collinear. In any case, I think some exploration of similar correlations to those shown in Fig. 4 would make the authors' case even stronger."*

We agree with the reviewer that this analysis would tie Figs 4 & 5 together nicely. However, the difference between attend-collinear and attend-orthogonal is only evident in an overall shift in the mean BOLD activity. Since we cannot compute a voxel-wise attention index (as a ratio) for Experiment 3, as individual responses can be negative, the Pearson correlation is not a good metric to reflect the degree of attentional enhancement.

- 3. "I'm not sure I'm convinced the authors have demonstrated that attention acts to reduce suppression induced by tuned normalization. As described above (Comment 2), the current model of attention acts on both the tuned and untuned normalization (and really, just acts as a general gain control in the numerator). To demonstrate that attention acts to reduce the impact of tuned normalization, the authors would need to demonstrate that attention primarily impacts the tuned normalization component of the model – that is, they'd need to move the $\lambda-1$ in front*

of the tuned component of Eq 5 (rather than distributing it across all normalization terms in the denominator). I'm not sure the present dataset would afford this test (not enough unique conditions tested). But, again, I may be missing something – I'm happy for the authors to correct me and clarify the manuscript as necessary to avoid reader misunderstanding.”

Although the model is no longer included in the manuscript, we have updated the language throughout the paper. The reviewer correctly points out that we currently can't say for certain that attention acts to reduce the impact of tuned normalization. Rather, that attention and tuned normalization appear to be related.

4. “Throughout the manuscript, the statistical comparisons could be improved. At present, there seems to be no correction for multiple comparisons when appropriate (e.g., across ROIs), and the authors use different metrics across experiments without justification (see Minor Point 6 below). There are also a few results with p 's hovering above 0.05 that are listed in support of a result (e.g., Fig. 5c, pg 10, V3). There are also cases where ANOVAs could be used to test for main effects/interaction, such as when comparing effect of stimulus configuration across ROIs.”

We have updated the statistical comparisons throughout the manuscript. We have performed repeated measures ANOVA's to consider changes across ROIs, and have applied corrections for multiple comparisons on post-hoc analyses.

5. “Fig. 1c: I'm not sure I understand the x/y axis labels. Is each label “orthogonal/collinear minus additivity”? If so, shouldn't most voxels be negative? It seems the point is to show that, voxel-by-voxel, response are more subadditive for the collinear configuration than the orthogonal configuration. Additionally, the description of this result at the end of the paragraph at the top of the next page suggests that whatever is labeled collinear should be lower than whatever is labeled orthogonal (consistent with Fig. 1a), yet this is not the immediate visual impression one gets from looking at Fig. 1c. Additionally, in keeping with the exceptionally transparent data presentation of Fig. 4, it would be useful to show this for the 3 ROIs “

We have fixed the mix-up of the axis labels in Figure 1c, and thank the reviewer for their sharp eye. Moreover, we now include the voxel-by-voxel subadditivity scatter plots for V2-V3 in the supplementary materials (Figure S8).

6. “Pg 6, the authors say “the magnitude of the feature-tuned aspect of normalization seemed to decrease in strength along the visual hierarchy”, but do not provide any statistical basis for this claim – if it's just a qualitative statement, specifying this would help (though if it could be statistically supported, that would be even better)”

As mentioned in response to comment 4, we have included appropriate statistical tests to support this claim: “While BOLD responses to either stimuli configuration across visual areas were fairly consistent in the degree to which they exhibited sub-additivity (untuned normalization), the magnitude of the feature-tuned aspect of normalization decreased in strength along the visual hierarchy (one-way ANOVA across visual areas linear term contrast $F(2,17) = 7.85, p = 0.005, \eta_p^2 = 0.511$)”, see page 6.

7. “Have the authors considered evaluating the reliability of the tuned normalization measure they compute per voxel? Perhaps something like split-half reliability, etc, could be useful to know

(especially if this measure is meant to be used as a potential proxy for impact of attention on neural responses)”

To assess the reliability of the tuned normalization, we computed the bootstrapped confidence interval for the top 25% voxels in each region of interest (1000 repetitions), which reveals that our voxel-wise measures of normalization are, by and large, quite stable. We now mention the reliability of this measure include in the results: “*Although there is heterogeneity in the magnitude of sub-additivity between voxels within a region, comparing the difference between the hypothetical additive sum with both collinear and orthogonal stimulus configurations revealed a consistent pattern, with the collinear configuration exhibiting larger sub-additivity compared to the orthogonal configuration –an effect that was highly reliable for the majority of voxels within V1 (Figure 1c, S3, & S4).*”, see page 6, and Figure S3.

8. “I’m also curious if the authors have evaluated the extent to which tuned normalization might vary across the visual field (and thus across the cortical surface) – is tuned normalization greater near the edges of the stimulus? The authors mention in the methods that they measure single-voxel receptive fields for most subjects, so it may be possible to examine this in the present dataset. Even looking at the amount of tuned normalization projected onto the cortical surface (and next to retinotopic mapping data) could provide hints for what types of mechanisms could be driving these effects and inspire future studies.”

Indeed, we have evaluated the extent to which tuned normalization varies across the visual field. There did not appear to be a significant change in tuned normalization across eccentricity, nor did there appear to be differences near edges of the stimulus. We have included these results in the manuscript: “*To explore the spatial distribution of these effects, we examined the correspondence between the retinotopic preference and attentional modulation or tuned normalization strength measures. Using pRF mapping procedures^{53,54}, we estimated the preference for spatial position for every voxel, allowing us to assess potential biases in both measures of interest, based on retinotopic preference. There did not appear to be any strong clustering of either of our two measures across spatial position. This suggests that while both attentional modulation and normalization strength share a tight relationship, this is not driven by a systematic coarse-scale topographical organization (see Figure S7).*”, see pages 8-9, and Figure S7.

9. “As a point of clarification – in Experiment 3 are both stimuli always presented with the same configuration on each trial?”

That is correct; during a trial both sides of fixation contained the same stimulus configuration. We have clarified this in the manuscript the Methods now says: “*Probes could appear on either side of fixation throughout a stimulus block, however observers were instructed to only respond to targets presented on the attended side, as indicated by the cue presented at fixation. The stimulus configuration displayed on either side of fixation was held constant within a block. Stimulus configuration and attention conditions were counter-balanced and presented in a pseudorandom order, and were interleaved with fixation blocks of equal duration.*” see page 20.

10. “Figs 3&4 and Fig. 5 use different metrics of signal modulation: Figs 3&4 (Experiments 1 & 2) compute a simple difference measure, while Fig. 5 (Experiment 3) uses a commonly-used

attentional modulation index measure (difference/sum). Either metric is fine, but it would be easiest to stick to one throughout the manuscript (or justify why different metrics are appropriate). On pg 22, they state that an AMI is “suggested to be a better representation when comparing an attentional effect between different conditions, as it is not biased by the differences in BOLD responses evoked by the two different stimuli configurations”, but provide no reference supporting this assertion. This may be helpful to include. If the authors decide to go with the AMI, it seems fair to also use a similar modulation index for the normalization metric. Note, of course, that modulation index measures are only useful for strictly-positive signals (like firing rates).”

As the reviewer notes, the ratio-based modulation index is only useful for strictly positive signals. Because we did have some voxels that exhibited negative responses, we were unable to calculate the AM as a ratio, and instead simply worked off the raw differences. We have now made the metric consistent throughout the manuscript and only report a difference to reflect attentional modulation.

11. “I overall really like Fig. 4. However, plotting data from the green subject up front and the blue subject in the back seems to highlight the strongest effect and hide the weakest one (based on the correlation values plotted in Fig. 4b). Could each participant be plotted individually in a supplementary figure? Or is there a way to convey this without essentially masking all but one subject?”

The ordering of plotting was not from strongest-to-weakest correlation, but instead we based it on order of data collection. That said, we agree it can be difficult to eyeball, and have provided a supplemental figure that shows the individual correlation plots, see Figure S8.

12. “Fig. S3: it seems that the x-axis of the bar graphs in panel b is mislabeled? Those seem to be conditions, while the dots are observers.”

The reviewer is correct, the x-axis is mislabeled. We have updated the figure and legend so that the dots are observers, and the x-axis represents the condition. The legend of Figure S9 now correctly describes the figure: “Stimuli are modified for illustrative purposes; dots indicate individual participants; error bars denote ± 1 S.E.M.”

13. “On pg 20 in the Methods, the authors disclose that behavioral performance was different between the attend-stimulus and attend-fixation tasks. Is this difference significant? Could behavioral performance differences account for any aspect of the results?”

For Experiment 2 we choose an all-or-none attention manipulation. While it is true that the fixation task is easier compared to the orientation discrimination task, this is unlikely to account for our results.

14. “For Experiment 3, what were the ‘probe’ stimuli? Neutral gray discs? And what was behavioral performance inside the scanner?”

The probe stimuli were 1.5 degree gray Gaussian discs, overlaid over the stimuli. We now include these details in the Methods. We have updated this information in the Results which now says: “Participants performed a demanding probe detection task, detecting and

discriminating whether a neutral gray Gaussian disk was embedded in the upper or lower visual field of the attended side of the screen, while maintaining fixation at the center of the screen (Figure 5a & S3, diameter probe 1.5°).”, see page 10. The methods now say: “Observers performed a demanding probe detection task, detecting and discriminating whether a neutral gray Gaussian disk appeared at a random location within the upper or lower visual field on the attended side of fixation (probe size 1.5°, with smoothed edges)”, see pages 19-20. Unfortunately, due to a programming oversight we were unable to recover participants performance inside the scanner, and therefore had them perform the task again outside of the scanner. Discrimination performance between the two configurations was comparable.

15. *The authors should include a data/code availability statement for the fMRI data and the computational models. I recommend posting data and code on a publicly-accessible repository, such as Open Science Framework and/or GitHub.*

We now include a data and code availability statement, and plan to make the data and code available via osf.io, see page 14.

Reviewer 3:

“This is an elegant and well-designed study that provides important new information about the mechanisms of attentional modulation, feature-based normalization, and their interaction. The correlation analysis shown in Figure 4 currently conflates between- and within-subject variation in attention modulation and tuned normalization strength, and this should be rectified. Also, adding results from a model comparison would strengthen the modeling findings. Minor issues and suggestions are listed below.”

We thank the reviewer for their kind words and comments. As mentioned above, we have removed the tuned normalization model from this manuscript, which accounted for a number of concerns that were raised. Outlined below, we have addressed the remainder of Reviewer 3’s concerns.

1. *“The correlation analyses presented in figure 4 are a combination of correlations due to individual differences in attention modulation and tuned normalization strength and those due to correlations of these measures across voxels in each subject. However, the authors’ interpretation is that the correlations are due to variability across voxels in each subject. The authors should recalculate the correlations after mean-centering the data for each participant.”*

The analysis presented in Figure 4 illustrates the relationship between attention modulation and tuned normalization across voxels within each participant. This relationship within each participant is quantified with a spearman correlation, which is not affected by the mean value of either measure. However, these spearman correlations cannot directly be compared between observers; therefore, we apply a Z transformation, allowing for a fair comparison between observers, we have clarified in the figure legend that these transformed Fisher-Z correlations are compared between observers, *“All Spearman correlations were transformed into a Fisher Z-statistic to allow for statistical comparisons between observers, see Methods.”*, see page 9. Furthermore, the Methods have been clarified and now says: *“To compare the degree of voxel-wise dependency between these two measures we computed a Spearman correlation, for the voxels that passed the stringent voxel selection described above within the defined V1-V3 regions, which was Fisher-Z transformed to allow for comparison between observers (Figure 4a)”*, see page 21.

2. *“Figures 1, 3, 4, and 5 present data from individual subjects in which each subject is assigned a color along a gradient (green to blue or dark gray to light gray). In my opinion, this data visualization is ineffective, as the colors representing the individual subjects are not distinguishable enough for the reader to be able to assess individual differences. Plotting the individual data points for each subject is commendable, but I think it would be best to use the same color for all participants.”*

While we understand the reviewer’s concern, we opted to use a consistent color coding for subjects, as it allows for better ability to distinguish between participants in Figure 4. We have included a supplementary figure (Fig S8), that shows the individual correlation plots.

3. *“The results of statistical tests comparing BOLD responses to orthogonal and collinear stimuli are presented twice, once on page 5 and again on page 7.”*

We have modified the text to only refer to these statistical tests once.

4. "page 9, "The less pronounced relationship in V3 is likely driven by the reduced heterogeneity and overall magnitude of tuned normalization strength observed in this visual area." Are the strength of correlation and the heterogeneity and magnitude of tuned normalization strength actually lower in V3 compared to V2?"

We have modified this sentence to reflect that the strength of the correlation is reduced in both V2 and V3. In the Results now says: "The less pronounced relationship in extrastriate cortex is likely driven by the reduced heterogeneity and overall magnitude of tuned normalization strength observed in these visual areas (**Figure 4 & S8**).", see page 10.

5. "page 8, Figure 4 legend, "A tight relationship between tuned normalization strength and attentional modulation is evident for the top 25% selected voxels for each observer." At this stage in the paper, it is unclear what is meant by "top 25% selected voxels", and this should be more clearly explained here."

The top 25% selected voxels were based on an independent functional localizer, for which we chose the top quarter of visually responsive voxels within our ROIs. We now expand on this in the main text of the Results, "Furthermore, to ensure that our results reflected a true relationship between normalization and attention, rather than being driven by differences in spurious factors, such as the signal-to-noise ratio (SNR), our analyses were restricted to the top 25% most visually responsive voxels within V1-V3, selected using an independent functional localizer", see page 9. Furthermore, the legend of Figure 4 now states: "A tight relationship between tuned normalization strength and attentional modulation is evident for the top 25% selected voxels for each observer, see Methods."

6. "page 13, "While the notion of feature-based attention is well established (59,60), incorporating this into the normalization model does not account for the results we have presented here." What is the evidence for this claim?"

We make this claim based on the results presented in Figure 2d & 5b. Here we showed there are differences between the collinear and orthogonal configurations in the absence of attention, as participants were either performing a demanding task at fixation finding targets in a rapidly presented letter stream (Experiment 2), or spatial attention was directed away from the stimulus (Experiment 3). We have clarified that we are referring to our experimental findings, "While the notion of feature-based attention is well established^{59,60}, incorporating this into the normalization model does not account for the results we have presented here. In our study, we provide evidence that differences between the two stimuli configurations emerge even when they were unattended (**Figure 2d & Figure 5b**), suggesting that tuned inhibition can arise in the absence of any aid of top-down attentional feedback.", see page 12.

7. "page 14, "While neuroimaging studies using a typical center-surround stimuli often report an attenuation of the response to the center stimulus..." One of the references listed here (Xing and Heeger, 2000) is not a neuroimaging study."

Thank you for spotting that omission. We have now changed the text so that it now says, "While neuroimaging and psychophysical studies using a typical center-surround stimuli often report an attenuation of the response to the center stimulus...", see page 13.

8. *“page 14, “We hypothesized that by keeping the orientation of one of the components constant and manipulating the orientation of the second component, we can induce normalization more analogous to surround suppression within all neural populations with receptive fields falling within our stimulus bounds. The superimposed configurations indeed elicited the predicted populations responses that one would expect from tuned normalization, as we found lower BOLD responses for those configurations that matched in their orientation content, compared to configurations with orthogonal orientation information.” Some references to the literature of orientation selectivity of surround suppression (both at the perceptual and neural level) would be helpful here.”*

We now include references to the literature related to perceptual and neural investigations of the feature tuned nature of surround suppression: *“We hypothesized that by keeping the orientation of one of the components constant and manipulating the orientation of the second component, we can induce normalization more analogous to surround suppression within all neural populations with receptive fields falling within our stimulus bounds. The superimposed configurations indeed elicited the predicted population responses that one would expect from feature tuned normalization^{15,65,68,69}, as we found lower BOLD responses for those configurations that matched in their orientation content, compared to configurations with orthogonal orientation information.”*, see page 13.

9. *“page 19, “A power analysis indicated that six participants would be sufficient to detect the reported normalization strength and attention effects.” More information should be provided about this power analysis. Did it incorporate variances and effect sizes taken from previous studies of normalization strength and attention effects?”*

Although there was very little prior fMRI work in tuned normalization, our estimate of power and number of subjects was based off of prior studies of human visual attention and vision (e.g. Gandhi et al., 1999; McMains and Somers, 2004; Herrmann et al., 2010; Jehee et al., 2011; Pestilli et al., 2011; Itthipuripat et al., 2014; Flevaris and Murray, 2015; Ling et al., 2015). We now include the following text in the Methods: *“A power analysis indicated that six subjects would be sufficient to detect the predicted attention effects. Indeed, this sample size is consistent with previous fMRI attention and vision studies⁷⁻⁸.”*, see page 18. Moreover, we have included appropriate effect size measures for all statistical analyses.

10. *“I don’t think that the properties of the probe used in the spatial attention experiment are described anywhere in the paper.”*

We have added details regarding the probe in both the Results and Methods to reflect that the probe was a gaussian neutral gray disk, subtending 1.5 degrees We have updated this information in the Results which now says: *“Participants performed a demanding probe detection task, detecting and discriminating whether a neutral gray Gaussian disk was embedded in the upper or lower visual field of the attended side of the screen, while maintaining fixation at the center of the screen (**Figure 5a & S9**, diameter probe 1.5°).”*, see page 10. The Methods now says: *“Observers performed a demanding probe detection task, detecting and discriminating whether a neutral gray Gaussian disk appeared at a random location within the upper or lower visual field on the attended side of fixation (probe size 1.5°, with smoothed edges)”*, see pages 19-20.

11. *“page 21, “Within the regions of interest, we defined the top 25% of voxels based on the independent localizer scans (using a standard GLM analysis) for those voxels whose estimated population receptive field (pRF) location fell within the stimulus aperture (15 degrees diameter).” How were the top 25% of voxels defined for those subjects who did not complete pRF scanning and instead participated in retinotopic mapping with periodic stimuli?”*

For the participant which did not complete pRF scanning we based our voxel cut-off on a fixed significance value from the GLM analysis. The methods now state: *“For the participant that did not complete pRF scanning, a fixed localizer significance cut-off was used that yielded a similar number of total voxels, compared to the other participants.”*, see page 21.

12. *“page 22, “This attention index is suggested to be a better representation when comparing an attentional effect between different conditions,...” Better than what?”*

We apologize for the ambiguity. Based on this and the other reviewer’s comments we have chosen to only report the attentional difference and not the ratio. We have modified Figure 5 and made changes throughout the Results and Methods to reflect this.

Reviewers' Comments:

Reviewer #1:

Remarks to the Author:

The authors have adequately addressed the concerns that I had from the previous round. The only further comment I have is that the introduction does not do justice to the existing literature on feature-based attention effects. First, the Reynolds & Heeger normalization model of attention explicitly has a section which simulates feature-based attention - so is not "feature-agnostic" as discussed in the intro. Second, feature-based attention effects have been studied in both human and animal studies which should both be cited (note that the sentence in the intro discussing feature-based attention has one animal citation #10, which does not seem to be in the bibliography).

Reviewer #2:

Remarks to the Author:

The authors have addressed all my concerns, and I congratulate them on an interesting and insightful study.

Reviewer #3:

Remarks to the Author:

General comments:

The authors have adequately responded to most of the issues I raised in the original review. Minor issues regarding writing and lack of clarity are listed below.

Specific comments:

Major points: None.

Minor points:

1) Reference #10 is missing from the References list.

2) Figure 1 legend, "...deviation from sub-additivity for both stimuli configurations..." should probably be "deviation from additivity for both stimulus configurations..."

3) page 6, "...we did not find a difference in the evoked responses to individual components based on orientation (45 degrees or 135 degrees; Figure S2)." This figure does not include a comparison of responses to the two orientations. I think it would be more accurate to state something like "...we did not find a difference in the evoked responses to radial versus tangential orientations..."

4) Figure S4: Is there any difference between collinear and orthogonal stimuli in the deviation from additivity in areas V2 and V3, or is this effect only evident in V1?

5) Figure S7: Panels c and d suggest that there is little or no effect of eccentricity on the strength of attentional modulation in V1-V3. However, Bressler et al. (2013) Vision Research showed strong eccentricity effects on attentional modulation of fMRI responses to visual stimulation in these areas. This should be discussed.

6) page 9, "...top 25% most visually responsive voxels within V1-V3,..." What does "most visually responsive" mean here? Largest response amplitude? Most reliable visual responses? Similar

comments for "Regions of interest" section of Methods and legends for Figure 4, S3 and S8.

7) page 12, "spatial extend" should be "spatial extent".

Michael Silver

Reviewer 1:

"The authors have adequately addressed the concerns that I had from the previous round. The only further comment I have is that the introduction does not do justice to the existing literature on feature-based attention effects. First, the Reynolds & Heeger normalization model of attention explicitly has a section which simulates feature-based attention - so is not "feature-agnostic" as discussed in the intro. Second, feature-based attention effects have been studied in both human and animal studies which should both be cited (note that the sentence in the intro discussing feature-based attention has one animal citation #10, which does not seem to be in the bibliography)."

We are pleased to see that we have adequately addressed most of the reviewer's comments. In this revision, we have modified our wording in the introduction to acknowledge that the normalization model of attention does incorporate feature-based attention. This paragraph in the Introduction now says: '*Normalization accounts of attention have traditionally hinged on three key components: the locus of attention, the size of the stimulus, and the size of the attentional window^{5,28,33}. However, while these models have proposed that the spatial extent of the 'attention field' could be feature selective^{5,26,56}, the suppressive drive itself is considered to be feature-agnostic, allowing an equal contribution of all information, regardless of feature similarities.*', see page 2.

Moreover, we further discuss the role of feature-based attention in the normalization model in more detail in the discussion: '*Previous work has suggested that instead of incorporating a tuned suppressive component into the normalization model of attention, a more parsimonious description to explain differences in the magnitude of suppression is by allowing attention to be feature-selective^{5,26,56}. While the notion of feature-based attention is well established^{68,59}, incorporating this into the normalization model does not account for the results we have presented here. In our study, we provide evidence that differences between the two stimuli configurations emerge even when they were unattended (**Figure 2d & Figure 5b**), suggesting that tuned inhibition can arise in the absence of any aid of top-down attentional feedback. Furthermore, in a third experiment we manipulated spatial attention, while holding factors such as stimulus size, attentional window size, and contrast relatively constant, in order to investigate the role that feature-tuned normalization has on attentional modulation (**Figure 5**). We find that the magnitude of attention is greatest when stimuli match in their visual features. This finding is not readily explained by the*

normalization model of attention, and instead would need to be extended such that feature similarity results in a release from normalization with attention., see page 11.

We thank the reviewer for spotting the missing reference in the bibliography, the Reference list has now been modified to include this reference.

Reviewer 2:

"The authors have addressed all my concerns, and I congratulate them on an interesting and insightful study."

We are pleased to see that we have adequately addressed all the reviewer's comments. We appreciate your time and effort in considering our work.

Reviewer 3:

"The authors have adequately responded to most of the issues I raised in the original review. Minor issues regarding writing and lack of clarity are listed below."

We are pleased to see that we have adequately addressed most of the reviewer's comments. Our replies to the remaining minor concerns are below.

1. *"Reference #10 is missing from the References list."*

The reviewer is correct, the reference was accidently omitted. The Reference list is now adjusted to include this reference.

2. *"Figure 1 legend, "...deviation from sub-additivity for both stimuli configurations..." should probably be "deviation from additivity for both stimulus configurations..." ."*

We have changed our wording in the legend of Figure 1: *"Voxel-wise relationship of the deviation from additivity for both stimuli configurations in V1"*, see page 4. As well as in Supplementary Figure 4, see Supplementary Information.

3. *“page 6, “...we did not find a difference in the evoked responses to individual components based on orientation (45 degrees or 135 degrees; Figure S2).” This figure does not include a comparison of responses to the two orientations. I think it would be more accurate to state something like “...we did not find a difference in the evoked responses to radial versus tangential orientations...” .”*

We agree with the reviewer that this is a better description for the radial bias analysis we report in Supplementary Figure 2. We have modified the manuscript: *'Note, we did not find a difference in the evoked responses to individual components when oriented radial or tangential (45° or 135°; **Supplementary Figure 2**)'*, see page 5.

4. *“Figure S4: Is there any difference between collinear and orthogonal stimuli in the deviation from additivity in areas V2 and V3, or is this effect only evident in V1?”*

We agree that it was hard to estimate the magnitude of the deviation from additivity for areas V2 and V3. To clarify this, we have added a new panel to the figure, which shows the proportion of voxels that fall below unity. Furthermore, we have made some adjustments to the main manuscript text: *'Although there was heterogeneity in the magnitude of sub-additivity between voxels within a region, comparing the difference between the hypothetical additive sum with both collinear and orthogonal stimulus configurations revealed a consistent pattern, with the collinear configuration exhibiting larger sub-additivity compared to the orthogonal configuration –an effect that was highly reliable for the majority of voxels within V1, and grew smaller for V2, and V3 (**Figure 1c, Supplementary Figure 3 & 4**)'*, see page 5. As well as in the Supplementary Figure legend: *'**Supplementary Figure 4. Voxel-wise relationship of the deviation from additivity for both stimuli configurations in V1-V3. BOLD responses were normalized for each participant. The average response for each participant demonstrated robust feature-tuned normalization, whereby stimuli comprised of collinear orientations were more sub-additive, and thus more strongly normalized, than stimuli that contained orthogonal orientations (repeated measures ANOVA interaction effect $F(2,15) = 7.85, p = 0.005, \eta_p^2 = 0.511$; tuned normalization post-hoc analysis two-sided paired t-test; V1: $t(5) = 6.00, p = 0.0057, d = 2.44$, V2: $t(5) = 3.82, p = 0.0370, d = 1.56$, and V3: $t(5) = 3.44, p = 0.0551, d = 1.42$, Bonferroni corrected). Individual voxel responses reveal large heterogeneity in the magnitude of sub-additivity within a region. Comparing the***

difference between the hypothetical additive sum with both collinear and orthogonal stimulus configurations revealed a consistent pattern, with the collinear configuration exhibiting larger sub-additivity compared to the orthogonal configuration – an effect that decreases in magnitude along the visual hierarchy. Small dots indicate individual voxels for each observer; larger black dots represent the whole ROI average per observer; N = 6.

5. *“Figure S7: Panels c and d suggest that there is little or no effect of eccentricity on the strength of attentional modulation in V1-V3. However, Bressler et al. (2013) Vision Research showed strong eccentricity effects on attentional modulation of fMRI responses to visual stimulation in these areas. This should be discussed.”*

We discuss the potential reason why our results do not show an eccentricity effect on attention modulation, while Bressler et al. (2013) does. The legend for Supplementary Figure 7 now says: *‘In contrast with previous work, we do not find an increase in attention modulation with eccentricity². This is likely due to differences in stimulus configuration, here participants attended a full field stimulus, which might explain why the magnitude of attention is consistent across eccentricity.’*

6. *“page 9, “...top 25% most visually responsive voxels within V1-V3...” What does “most visually responsive” mean here? Largest response amplitude? Most reliable visual responses? Similar comments for “Regions of interest” section of Methods and legends for Figure 4, S3 and S8.”*

We apologize for the ambiguity of the description of our voxel selection. We have modified the text in the manuscript: *‘our analyses were restricted to the top 25% most visually responsive voxels within V1-V3, selected by ranking the significance obtained from a standard GLM analysis of an independent functional localizer.’*, see page 8. As well as in the Methods: *‘Within the regions of interest, we defined the top 25% of voxels based on the independent localizer scans (using a standard GLM analysis, selecting the most visually responsive voxels by their respective significance values) for those voxels whose estimated population receptive field (pRF) location fell within the stimulus aperture (15° diameter).’*, see pages 15-16.

7. *"page 12, "spatial extend" should be "spatial extent" ."*

We thank the reviewer for their sharp eye. This typo has now been corrected.